# Intermediately synchronised brain states optimise trade-off between subject specificity and predictive capacity

Leonard Sasse [1,2,3], Daouia I. Larabi [1,2], Amir Omidvarnia [1,2], Kyesam Jung [1,2], Felix Hoffstaedter [1,2], Gerhard Jocham[4], Simon B. Eickhoff[1,2] & Kaustubh R. Patil [1,2 ✉]

Functional connectivity (FC) refers to the statistical dependencies between activity of distinct brain areas. To study temporal fluctuations in FC within the duration of a functional magnetic resonance imaging (fMRI) scanning session, researchers have proposed the computation of an edge time series (ETS) and their derivatives. Evidence suggests that FC is driven by a few time points of high-amplitude co-fluctuation (HACF) in the ETS, which may also contribute disproportionately to interindividual differences. However, it remains unclear to what degree different time points actually contribute to brain-behaviour associations. Here, we systematically evaluate this question by assessing the predictive utility of FC estimates at different levels of co-fluctuation using machine learning (ML) approaches. We demonstrate that time points of lower and intermediate co-fluctuation levels provide overall highest subject specificity as well as highest predictive capacity of individual-level phenotypes.

---

[1] Institute of Neuroscience and Medicine, Brain and Behaviour (INM-7), Research Centre Jülich, Jülich, Germany. [2] Institute of Systems Neuroscience, Medical Faculty, Heinrich-Heine-University Düsseldorf, Düsseldorf, Germany. [3] Max Planck School of Cognition, Stephanstrasse 1a, Leipzig, Germany. [4] Institute for Experimental Psychology, Faculty of Mathematics and Natural Sciences, Heinrich-Heine-University Düsseldorf, Düsseldorf, Germany. ✉email: k.patil@fz-juelich.de

In an effort to understand how brain organisation facilitates flexible yet specialised cognitive function, much neuroscientific research has focused on the functional connectivity (FC) between brain areas by investigating their pairwise correlations of functional magnetic resonance imaging (fMRI) blood oxygen level-dependent (BOLD) signals[1–3]. These pairwise correlations are assumed to represent the strength of connectivity (also called edges) between brain areas (also called nodes). Notably, considerable promise for the eventual applicability of FC as a biomarker has been shown with numerous studies demonstrating that FC differs between individuals, is stable within an individual[4–8], and relates to individual-level cognition[4,9–13] as well as clinically relevant symptoms of mental disorders[14–17]. In this context, recent research has focused on investigating how different time points in the rs-fMRI time series contribute to some of these properties of FC[18–20]. However, it is not yet well understood how different time points contribute specifically to the predictive utility of FC.

In order to move closer to the goal of applicability of FC biomarkers for real-world applications, researchers have tried to improve behavioural prediction by searching for the most suitable feature engineering schemes[9], machine learning algorithms[10,12], preprocessing parameters and brain parcellations[21,22]. Others focused on optimisation of within-individual stability and uniqueness of the FC fingerprint. This can be investigated within the identification framework in which the success of this optimisation is reflected in a higher success rate for the identification of an individual based on their FC profile or FC fingerprint[4]. With one such promising approach edge time series (ETS) are leveraged to select specific time points to estimate FC[18]. These time points of high-amplitude co-fluctuations, called events, contribute disproportionately to FC and are thought to reflect fluctuations in cognitive state[18]. Despite the high magnitude of changes in resting state fMRI signals, the extent to which they contribute to cognition and behaviour is still a matter of debate in the literature[23–25]. Although approaches to decompose task activity into underlying recruitment of resting state networks have been proposed, further research is needed to investigate how resting state signals and their changes over time underlie cognitive and behavioural performance[26].

These ETS reflect the magnitude of co-fluctuations of each pair of brain areas over time[27,28] and are calculated as the product between the z-scored time series for each pair of brain areas. To study overall co-fluctuation patterns across all areas, the root sum of squares (RSS) at each time point along the ETS has been suggested as a meaningful measure of co-fluctuation amplitude[18]. A higher RSS indicates higher co-fluctuations (or ETS) across the brain, i.e. higher overall brain synchronisation. This allows for the selection of only high-amplitude co-fluctuation (HACF) or low-amplitude co-fluctuation (LACF) time points from the original BOLD time series. HACF-derived FC has been shown to yield enhanced subject specificity, bringing up the question whether using these HACF time points might also amplify brain-behaviour correlations[18,19,29].

Thus far, a crucial missing component in the investigation of ETS is the evaluation of individual differences at different co-fluctuation amplitudes by means of prediction of phenotypes. Previous research has shown that connectivity of brain areas that contribute most to identification accuracy do not overlap with brain areas that contribute most to prediction accuracy[30]. This suggests that FC uniqueness and stability on their own do not guarantee phenotypic relevance of brain connectivity representations[31]. Furthermore, it is possible that different subsets of time points are more or less predictive of different phenotypic domains. In addition, one may ask, whether any selected subset of time points is more predictive than the full FC estimated using all available time points. Previous research has suggested that, to some degree, distinct brain states are differently associated to specific behaviours. For example, brain states showing strong integration between functional networks are associated with better cognitive task performance, in particular for memory- and attention-related tasks[32–34]. In addition, brain states characterised by high modularity are associated with better performance during motor tasks[35,36].

Therefore, to address these questions, we systematically investigated the phenotypic relevance of FC contributions at different amplitudes of ETS co-fluctuations. Employing three time point sampling strategies, we investigated predictiveness of targets across the domains of cognition, behaviour, personality, and demographics from FC including all time points, HACF time points, LACF time points, or a combination thereof using the Human Connectome Project (HCP-YA) S1200 dataset[37,38]. We also validate our findings using the Human Connectome Project Aging (HCP-A) dataset[39,40]. Further, to get a better understanding of what drives co-fluctuations, we investigated the influence of structural connectivity on the relationship between FC estimates at different co-fluctuation levels and predictiveness of these targets.

## Results

For each of the four resting state fMRI (rs-fMRI) runs of 771 subjects of the HCP-YA S1200 dataset[37,38], we computed the edge time series (ETS) and the corresponding root sum of squares (RSS) along the ETS to quantify co-fluctuation amplitude using the Schaefer parcellation with 200 parcels[41]. For each subject, BOLD time series were ordered ranging from time points with high overall co-fluctuation across the brain (high RSS; high amplitude co-fluctuation [HACF]) to low overall co-fluctuation (low amplitude co-fluctuation [LACF]). Next, we used three different strategies for selection of time points with different levels of co-fluctuation: (1) sequential sampling: consecutively includes differing percentages of HACF or LACF using a threshold ranging from 0 to 50%, (2) individual bins: all time points were divided into 20 individual bins each comprising 5% of time points, and (3) combined bins: included time points of all possible combinations of individual bins (see Methods, subsection Edge Time Series Construction and Functional Connectivity Estimation, and Supplementary Fig. 1 for details). FC matrices were created by calculating Pearson correlation coefficients between time series of each pair of brain areas while only including the selected time points. Using these distinct FC matrices allowed us to systematically investigate the contributions of time points with different levels of co-fluctuation to subject specificity and prediction of 25 phenotypes. Identification was performed between rs-fMRI data from day 1 and day 2 of HCP-YA data collection and for each day FC was averaged across phase encoding directions. For prediction, the FC matrices obtained on both days were averaged, resulting in one FC matrix per subject per co-fluctuation bin.

**Differential identifiability and identification accuracy disagree in their assessment of functional connectivity fingerprints**. To assess variance in identification accuracy and differential identifiability, we resampled the original 771 subjects 1000 times with replacement, keeping only the unique subjects from the resampled subject list. In the sequential sampling strategy, we could replicate the finding that HACF time points yield higher differential identifiability ($I_{Diff}$) than LACF time points (Fig. 1a). To assess statistical significance of this observation we used two-tailed Wilcoxon-signed-rank tests with a Bonferroni correction for multiple comparisons. Across all specified sampling

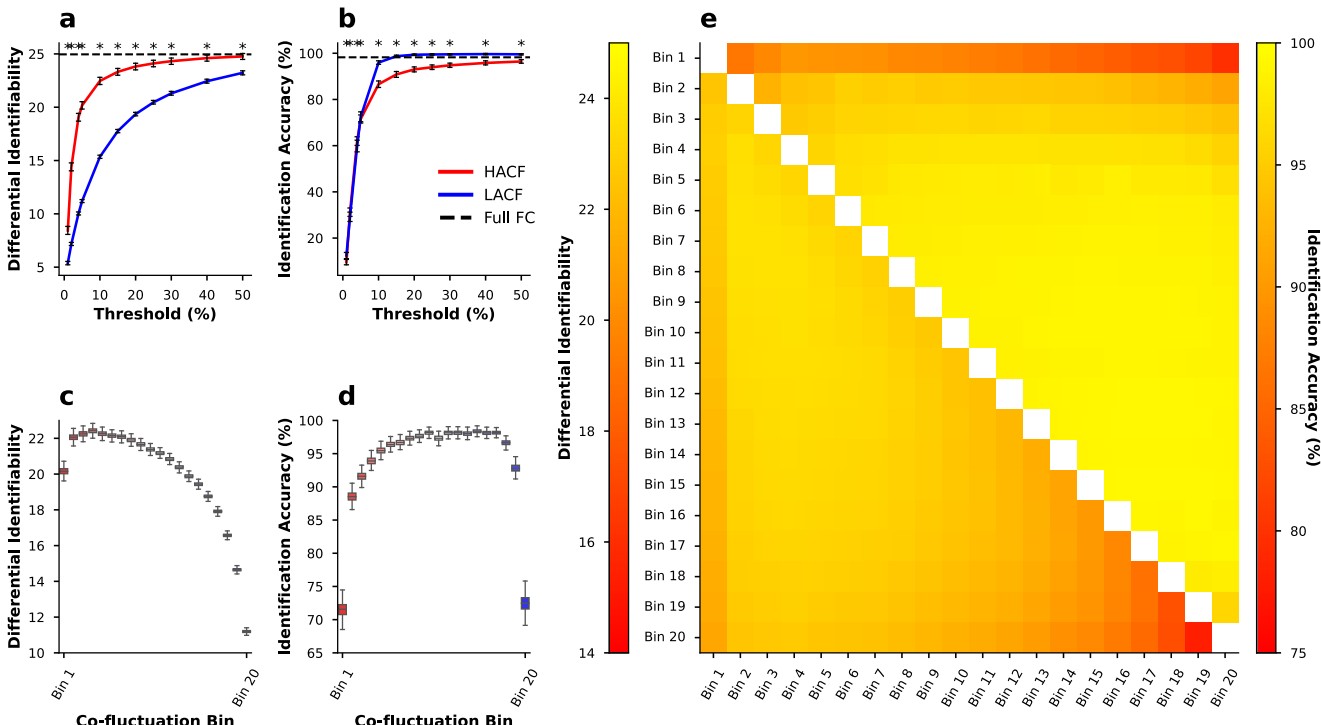

**Fig. 1 Subject specificity in the HCP-YA dataset assessed using two metrics and three different sampling strategies.** Differential identifiability and identification accuracy are obtained for the sequential sampling strategy (**a**, **b**), the individual bins sampling strategy (**c**, **d**), and the combined bins strategy (**e**). In (**a**, **b**), threshold refers to the percentage of highest (HACF) or lowest (LACF) co-fluctuation time points chosen to estimate FC. In (**e**), the lower triangle shows differential identifiability, whereas the upper triangle shows identification accuracy achieved by each pair of combined bins. In each identification experiment subjects were resampled with replacement 1000 times, and in each resampling run only the subset of unique subjects were chosen to perform the identification analysis. Asterisks (*) in (**a**, **b**) indicate a significant difference as determined by a Wilcoxon-signed rank test between HACF-derived FC and LACF-derived FC at the .05 alpha significance level after Bonferroni correction and the error bars indicate a 95% confidence interval. In (**c**, **d**) the box plot indicates the median (centre line) and the interquartile range. Source data can be obtained from Supplementary Data 1.

thresholds, HACF-derived FC provided statistically significant higher differential identifiability scores at the .05 alpha level of significance (Fig. 1a). At the same time, the sequential sampling strategy shows that LACF time points provide statistically significantly higher identification accuracy ($I_{Acc}$) than HACF time points across all specified sampling thresholds (Fig. 1b). Moreover, it becomes evident across the individual and combined bins sampling strategies that the highest $I_{Diff}$ is in fact achieved by intermediate bins (Fig. 1c). In the individual and combined bins sampling strategies, bins of intermediate co-fluctuation achieved the highest $I_{Acc}$ overall (Fig. 1d, e). In the sequential sampling strategy, it is most apparent that $I_{Acc}$ and $I_{Diff}$ show opposing effects (Fig. 1a, b).

**Functional connectivity estimated at intermediate levels of co-fluctuation yield higher prediction accuracy than high-amplitude co-fluctuation or low-amplitude co-fluctuation time points.** To test whether the differing behaviour of $I_{Diff}$ and $I_{Acc}$ can inform about the predictive utility, we then applied the sampling strategies to predict phenotypes using kernel ridge regression. We selected phenotypes used in ref. [42] from the categories Cognition, In-scanner task performance, and Personality (see Table S1). Results displayed here consist of the 9 phenotypic targets with the highest prediction accuracy based on the full FC. We report results for other difficult-to-predict targets (Supplementary Figs. 2 and 3) as well as results using the coefficient of determination ($R^2$) as scoring metric (Supplementary Figs. 4 and 5) in the supplementary information. To assess the out-of-sample prediction accuracy of our models in the HCP-YA dataset, we performed a 10-fold nested cross-validation (CV)

procedure. We ensured that family members were always kept within the same fold in the 10-fold CV to maintain independence between folds. We used a 5-fold inner CV on the training folds to select the hyperparameters for ridge regression (l2-regularisation strength) in a CV-consistent manner. The best parameters were then fitted on the training folds of the outer CV and tested on the outer CV test fold. Similarly, to avoid test-to-train leakage during confound removal, we trained a confound regression model on the training data only, to remove the effects of age, sex, and framewise displacement (for more information see Methods, subsection Prediction of Behavioural and Demographic Measures).

In the individual and combined bins sampling strategies a general trend can be observed with LACF time points yielding higher prediction scores than HACF time points (Fig. 2). Using a 5% Bayesian ROPE[43] (see Methods, subsection Prediction of Behavioural and Demographic Measures) to compare each co-fluctuation bin's performance against the performance of full FC, we found that across most targets co-fluctuation bins show predictive utility that is equivalent to the full FC (Fig. 2). However, in particular for the two targets that overall can be predicted with the highest accuracy (Reading and Vocabulary), it is apparent that HACF bins actually yield meaningfully lower prediction accuracy scores than full FC.

In the sequential sampling strategy LACF time points also consistently yielded better prediction accuracy than HACF time points (Fig. 3). Results were consistent across scoring metrics. To illustrate robustness, we performed several additional analyses with different settings and parameters. To this end, we repeated the same analysis using Connectome-based Predictive Modelling

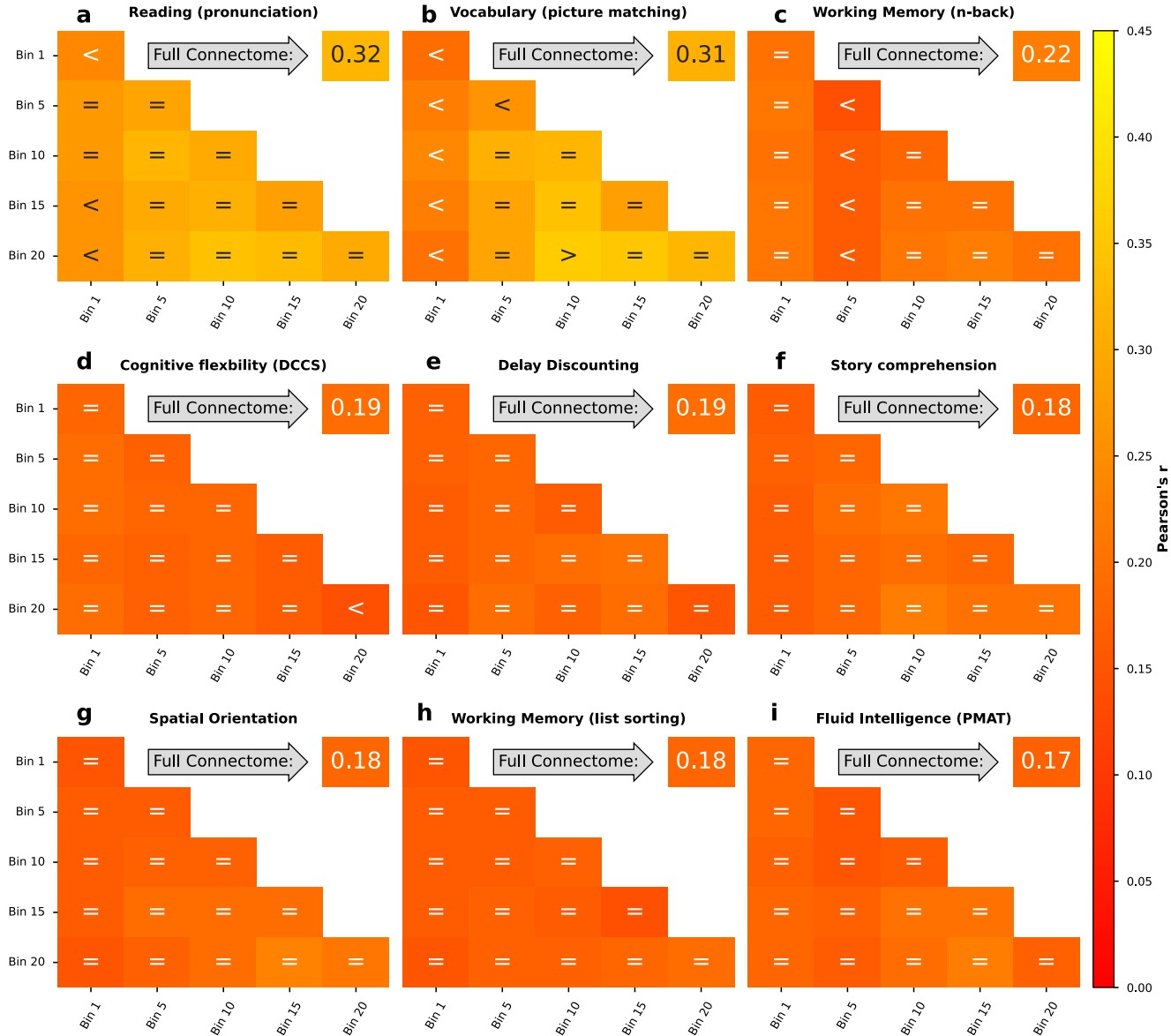

**Fig. 2 Prediction scores (Pearson's r between observed and predicted values) for 9 phenotypic targets averaged across the ten folds in the grouped cross-validation scheme when using combined and individual bins sampling strategies.** Each panel (**a–i**) shows results for a different target. Bins range from 1 (HACF) to 20 (LACF). Scores for individual bins are displayed on the diagonal, for combined bins off the diagonal. Scores for the full FC using the whole time series are always displayed in the upper right corner. Comparison operators indicate whether scores obtained by a co-fluctuation bin are equivalent to scores obtained by full FC ("=") or whether they are less ("<'') or greater (">'') than scores obtained by full FC according to a 5% Bayesian ROPE[43]. These 9 targets are displayed, because they yielded best prediction accuracy using full FC compared to other targets displayed in Supplementary Fig. 2. Source data can be obtained from Supplementary Data 2.

(CBPM)[9]. Using this approach, prediction scores were lower, but overall a similar pattern was observed (see Supplementary Figs. 6–9). We further provide results for a subset of targets using the Schaefer 300 and 400 parcellations, with or without global signal regression, and for both scoring metrics (Pearson's r and $R^2$), as these processing and analysis choices have been shown to affect prediction accuracy[22] (Supplementary Figs. 10–19). For this analysis we chose Reading and Vocabulary, because they yielded the best prediction accuracy scores as well as Fluid Intelligence, because it is widely used for prediction in the literature[4,44].

Overall, we made two robust observations across all analyses; LACF time points provide better predictive power than HACF time points, and some intermediate time points perform even better. To test whether these observations may be due to a

relationship between the RSS and in-scanner motion, we correlated the RSS time series with framewise displacement (FD) for every subject and every rs-fMRI run. These correlations follow a normal distribution centered at zero and therefore show little evidence of a relationship between RSS and in-scanner motion (Supplementary Fig. 20).

Given the importance of inter-individual demographic differences in basic and clinical research, we also tested whether this effect can be found when predicting age and sex which have shown better prediction accuracy than psychometric variables using FC[12,45]. In sex classification, we use a ridge classifier. In fact, a pattern similar to the previous analysis can be observed for both age and sex. In age prediction, the sequential sampling strategy clearly shows that HACF-derived FC yields lower prediction scores than LACF-derived FC (Fig. 4a). LACF-

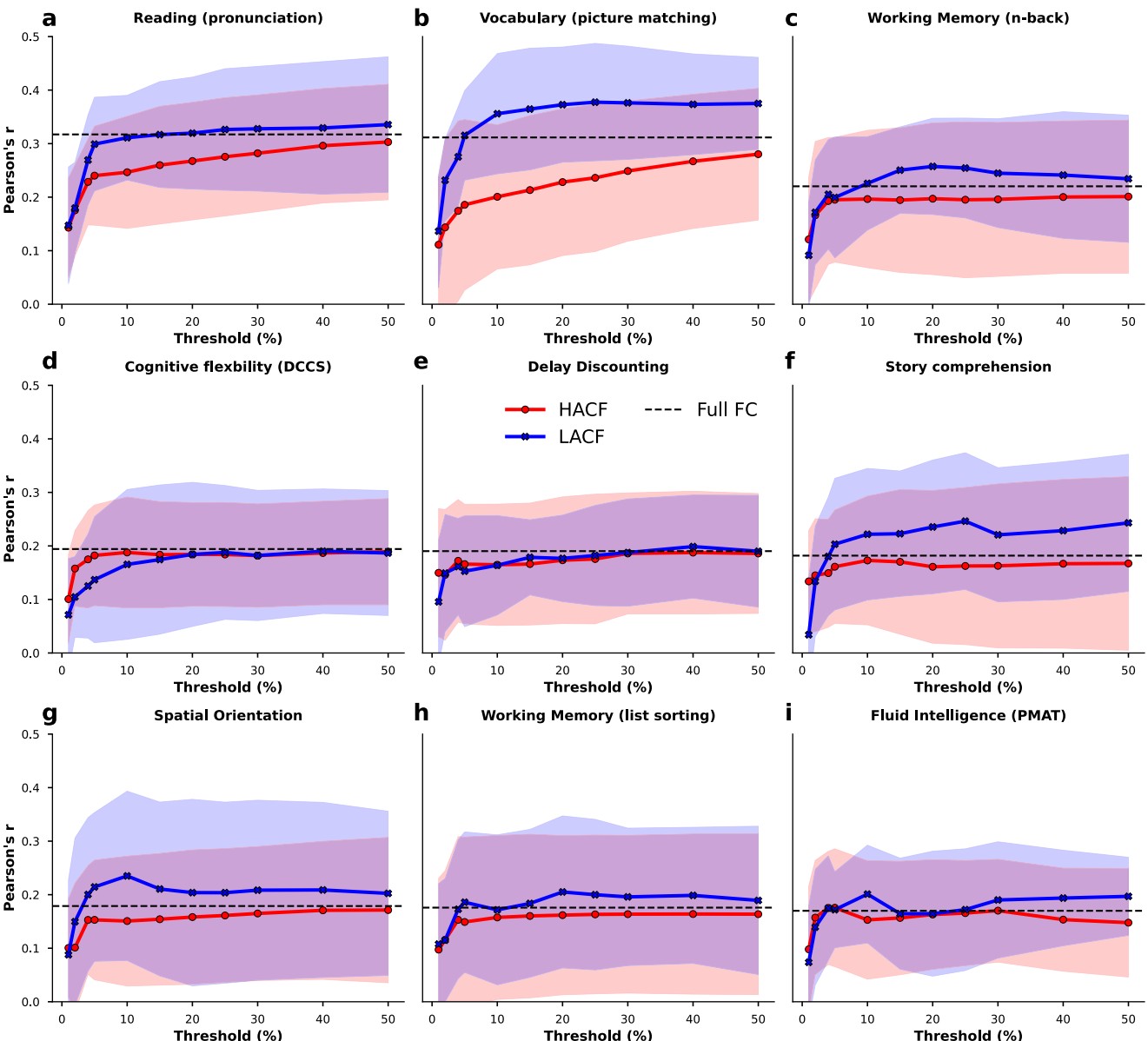

**Fig. 3 Prediction scores (Pearson's r between observed and predicted) for 9 phenotypic targets averaged across the ten folds in the grouped cross-validation scheme when using FC estimates derived from time points at different levels of co-fluctuation magnitude using the sequential sampling strategy.** Each panel (**a–i**) shows results for a different target. Threshold refers to the percentage of highest (HACF) or lowest (LACF) co-fluctuation time points chosen to estimate FC. Upper and lower boundary of the fill colours indicate the standard deviation across folds. A threshold of 100% corresponds to full FC. These 9 targets are displayed, because they yielded best prediction accuracy using full FC compared to other targets displayed in the Supplementary Fig. 3. Source data can be obtained from Supplementary Data 2.

derived FC even yields higher prediction scores than the full FC, a pattern that can be found also in the combined bins strategy (Fig. 4e). In the individual bins strategy it is further evident that intermediate bins typically yield higher prediction scores than both HACF and LACF time points (Fig. 4c). We further report age prediction results using $R^2$ (Supplementary Fig. 21) and mean absolute error (MAE - Supplementary Fig. 22) as scoring metrics. Considering the limited age range (22-37 years) in this sample, the age prediction scores are reasonable.

In sex classification, combined bins largely yielded prediction scores equivalent to full FC, however, a trend of prediction scores decreasing with higher co-fluctuation levels can be observed (Fig. 4e). In the individual bins strategy, both HACF and LACF bins yielded lower scores than intermediate bins (Fig. 4d). Lastly, comparing HACF and LACF time points using the sequential

strategy, it can be seen that LACF time points usually provide better prediction scores than HACF time points (Fig. 4b). Again, to test whether results are robust across different models, we repeated sex prediction using support vector classifiers (SVC) with a linear kernel (Supplementary Fig. 21) as well as a radial basis function (RBF) kernel (Supplementary Fig. 22). The results were very similar.

**Validation dataset yields similar results**. We then attempted to replicate identification and prediction in the HCP Aging (HCP-A) dataset (see Methods, subsection Datasets). Results for $I_{Acc}$ and $I_{Diff}$ in the HCP-A dataset were consistent with the findings in the HCP-YA dataset across all sampling strategies (Fig. 5). $I_{Acc}$ is higher for LACF time points than for HACF time points, whereas

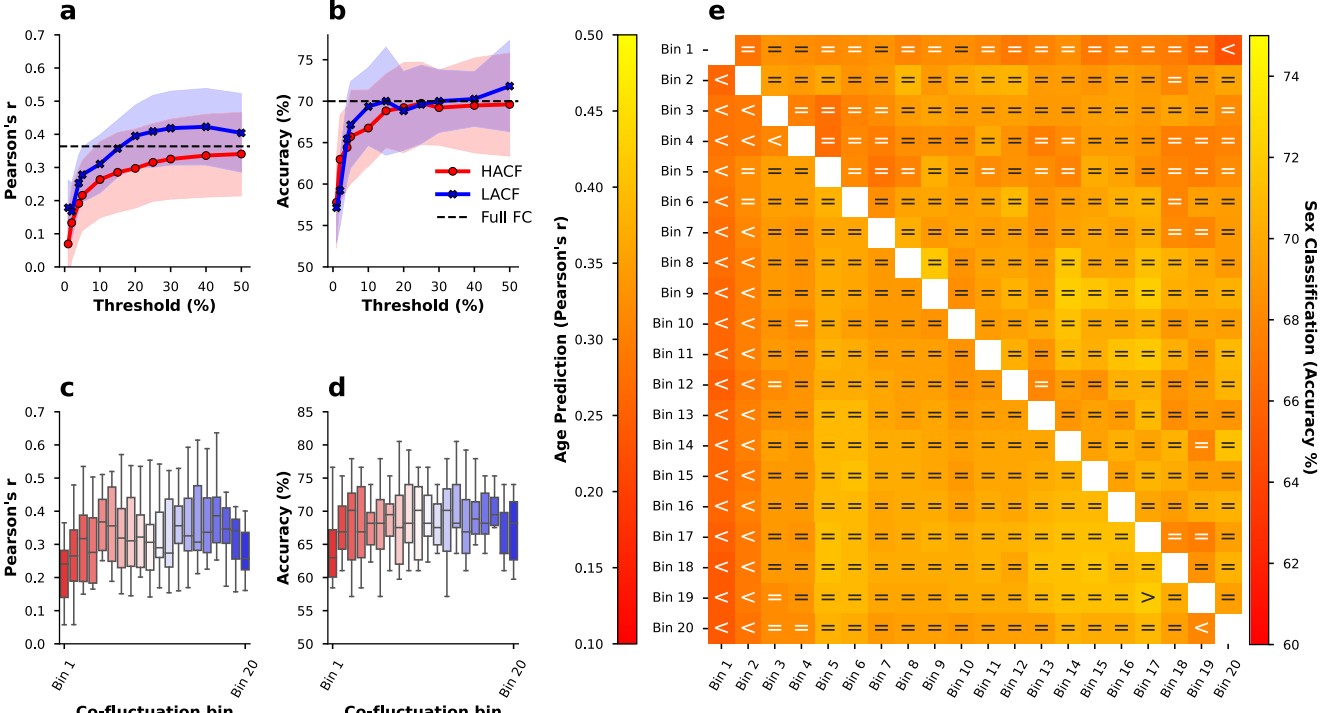

**Fig. 4 Age prediction scores (Pearson's r) and sex classification scores (accuracy) using FC at different levels of co-fluctuation.** Results are shown for the sequential sampling strategy (**a**, **b**), individual bins (**c**, **d**), and combined bins (**e**) sampling strategies. In (**a**, **b**), threshold (x-axis) refers to the percentage of highest (HACF) or lowest (LACF) co-fluctuation time points chosen to estimate FC and the y-axis indicates the prediction score. Upper and lower boundary of the fill colours indicate the standard deviation across folds. In (**c**, **d**) the box plot indicates the median (centre line) and the interquartile range. In (**e**) comparison operators indicate whether scores obtained by a co-fluctuation bin are equivalent to scores obtained by full FC (=) or whether they are less (<) or greater (>) than scores obtained by full FC according to a 5% Bayesian ROPE[43]. Source data can be obtained from Supplementary Data 3.

$I_{Diff}$ is higher for HACF time points than for LACF time points, indicating that this effect is not sample-specific.

For prediction in the HCP-A sample we selected Language/Vocabulary Comprehension and Cognitive Flexibility since these measures were also available in the HCP-YA sample and showed reasonable prediction accuracy. As there was no direct test of fluid intelligence in the HCP-A sample, we also included composite measures of fluid and crystallised cognition (see Table S2). Results on four cognitive targets show that prediction scores for these targets are overall largely equivalent to the full FC (Fig. 6a–h). However, a trend can be observed such that intermediate bins yield slightly higher performance than HACF or LACF time points. Similar results are obtained using $R^2$ as scoring metric (Supplementary Fig. 23). Next, we also set out to predict age and sex in the HCP-A dataset. However, since this sample was not balanced (316 female, 242 male), for sex classification we present balanced accuracy as scoring metric[46]. Results showed a similar pattern as previously observed but note the higher prediction accuracy for age on this sample due to its wider age range (Fig. 6i, k, m). In the individual and combined bins sampling strategies (Fig. 6k, l, m) scores are largely considered equivalent to prediction using the full FC, with a trend hinting at better performance in intermediate bins compared to HACF or LACF bins. Further, the sequential sampling strategy (Fig. 6i, j) makes it apparent that again LACF time points consistently yield better predictive performance compared to HACF time points. Again, in the supplementary material we report results for age prediction using $R^2$ (Supplementary Fig. 24) and MAE (Supplementary Fig. 25) as well as results for sex prediction using a SVC with linear kernel (Supplementary Fig. 24) and a RBF kernel (Supplementary Fig. 25). Results for these parameters were consistent with the results displayed here.

**Functional connectivity shows a stronger relationship to structural connectivity during intermediate levels of co-fluctuation.** As a last analysis, we aimed to investigate the correspondence between FC at different levels of co-fluctuation and structural connectivity. As a first step, we correlated each subject's SC with each co-fluctuation bin FC estimate. Our findings indicate that overall correlations between SC and FC tend to be greater for intermediate bins and some LACF bins compared to HACF bins (Fig. 7).

This trend is true for each of the three sampling strategies used. After obtaining these results we further wanted to test whether the information on underlying SC present in the intermediate bins and LACF bins indeed relates to the greater predictive utility of these bins (as compared to HACF bins). To test this we regressed out SC from the FC estimates for each subject and each co-fluctuation bin using a linear model and used the residuals for prediction of three cognitive variables in the HCP-YA dataset for which we observed a difference between HACF and LACF time points with respect to their predictive utility (Figs. 2, 3). The removal of SC from FC did not decidedly change prediction scores (Fig. 8a–f). We further repeated this paradigm in the prediction of sex and age. Here, the expected finding can be observed to a greater degree in the prediction of age (Fig. 8g, compare to Fig. 4a). Overall, however, regressing out SC does not seem to meaningfully change prediction scores, considering that the effect of intermediate time points and LACF time points yielding better predictions than HACF time points remains.

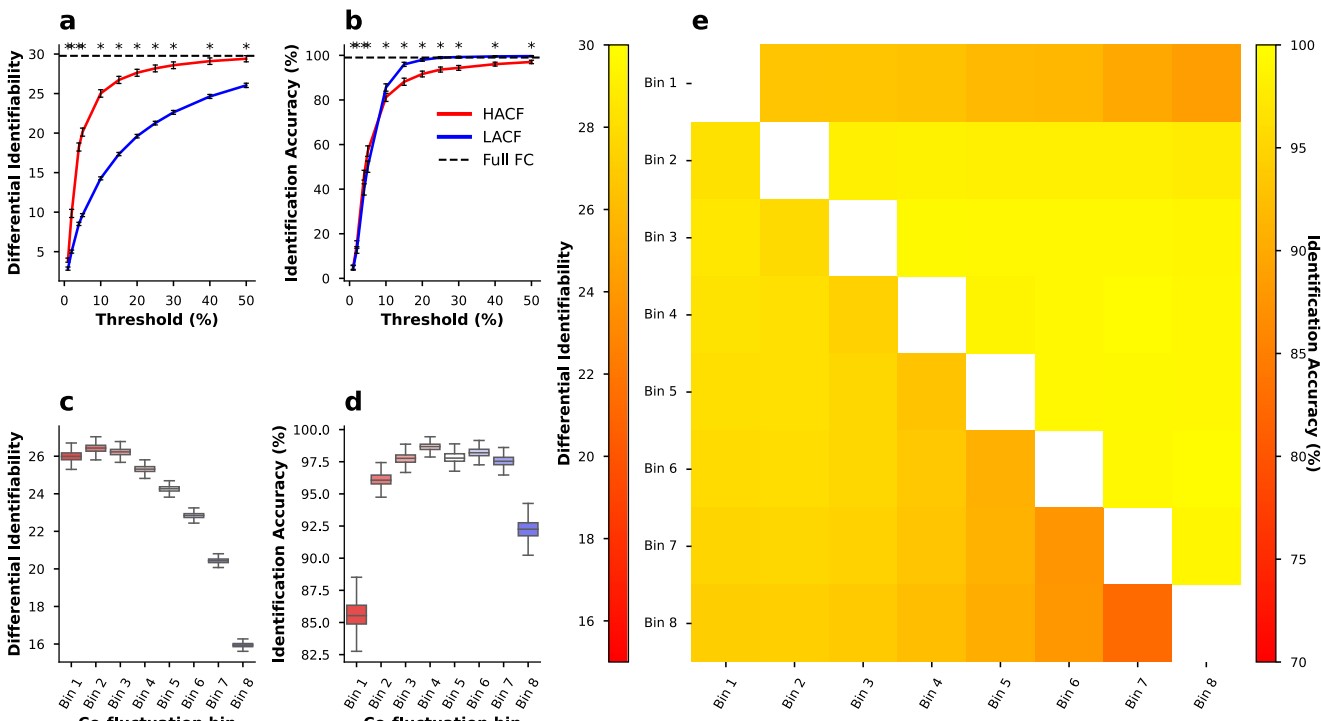

**Fig. 5 Subject specificity in the HCP-A dataset assessed using two metrics and three different sampling strategies.** Differential identifiability and identification accuracy are obtained for the sequential sampling strategy (**a**, **b**), the individual bins sampling strategy (**c**, **d**), and the combined bins strategy (**e**). In (**a**, **b**), Threshold (x-axis) refers to the percentage of highest (HACF) or lowest (LACF) co-fluctuation time points chosen to estimate FC and the y-axis displays the subject specificity ($I_{diff}$ or $I_{acc}$). In (**e**), the lower triangle shows differential identifiability, whereas the upper triangle shows identification accuracy achieved by each pair of combined bins. In each identification experiment subjects were resampled with replacement 1000 times, and in each resampling run only the subset of unique subjects were chosen to perform the identification analysis. Asterisks (*) in (**a**, **b**) indicate a significant difference as determined by a Wilcoxon-signed rank test between HACF-derived FC and LACF-derived FC at the .05 alpha significance level after Bonferroni correction. The error bars indicate a 95% confidence interval. In (**c**, **d**) the box plot indicates the median (centre line) and the interquartile range. Source data can be obtained from Supplementary Data 1.

## Discussion

It has been suggested that the use of HACF time points with enhanced subject specificity may amplify brain-behaviour associations[20,29]. On the other hand, it has also been shown that FC-based identification and prediction may constitute conflicting goals[30,31]. Therefore, here we systematically evaluated the effect of inclusion of varying levels of functional co-fluctuations on subject specificity and predictiveness of a range of phenotypes. Across a broad range of analytical settings and in two different cohorts, we observed that time points with *intermediate* levels of co-fluctuation yield highest subject specificity (i.e. highest identification accuracies and differential identifiability between two rs-fMRI sessions) and greater predictive power compared to the other time points. Altogether, our findings suggest time points with intermediate levels of co-fluctuation as a sweet spot capturing subject-specific phenotypic information; an intermediately synchronised and heterogeneous brain state situated in between a stereotypical highly synchronised brain state (HACF) and a weakly synchronised variable brain state (LACF). The highlighted role of intermediate co-fluctuation amplitudes in our study can be seen as a balanced point between increased synchrony (HACF) and increased disorder (LACF) between brain areas. Such a situation has also been reported at the brain neuronal level through the criticality hypothesis[47] where neuronal activity of brain circuits self-organises into critical states or a transition between order and disorder. It also implies that the most relevant information of large-scale BOLD dynamics is encoded through discrete events in contrast to the dominant view of continuous fMRI data analysis[48].

A previous study showed that differential identifiability of HACF-derived FC was higher than of LACF-derived FC[18]. Although we replicated this finding in our current study, we observed the opposite pattern for the identification problem with higher accuracies for LACF in contrast to HACF time points. This finding is in line with another study on individual differences in FC estimated at different levels of co-fluctuation[20], also showing that $I_{diff}$ was highest in HACF time points while $I_{acc}$ was highest at intermediate time points, and that removal of HACF time points decreased $I_{diff}$ while increasing $I_{Acc}$. Looking further into the subscores with which $I_{diff}$ is calculated (within- and between-subject correlations), it can be seen that for HACF time points the within-subject correlations increase more than the between-subject correlations (leading to higher $I_{diff}$; see Supplementary Figs. 26–28). This suggests a highly synchronised brain state that is stable within a subject. For LACF time points, on the other hand, within-subject correlations decrease more than between-subject correlations (leading to lower $I_{diff}$), suggesting this weakly synchronised brain state to be less stable within a subject. Integrating this with our identification accuracy findings, the lower identification accuracies for HACF suggest that this within-subject stable, highly synchronised brain state is also similar across subjects, i.e., a more stereotypical brain state[49]. Enhanced similarity between subjects at the HACF time points is also in line with point process analysis findings where the spatial maps of resting state networks can be reconstructed from a limited number of high-amplitude events of fMRI[48]. It may explain higher uniformity of HACF time points across the population. The higher identification accuracies for LACF may

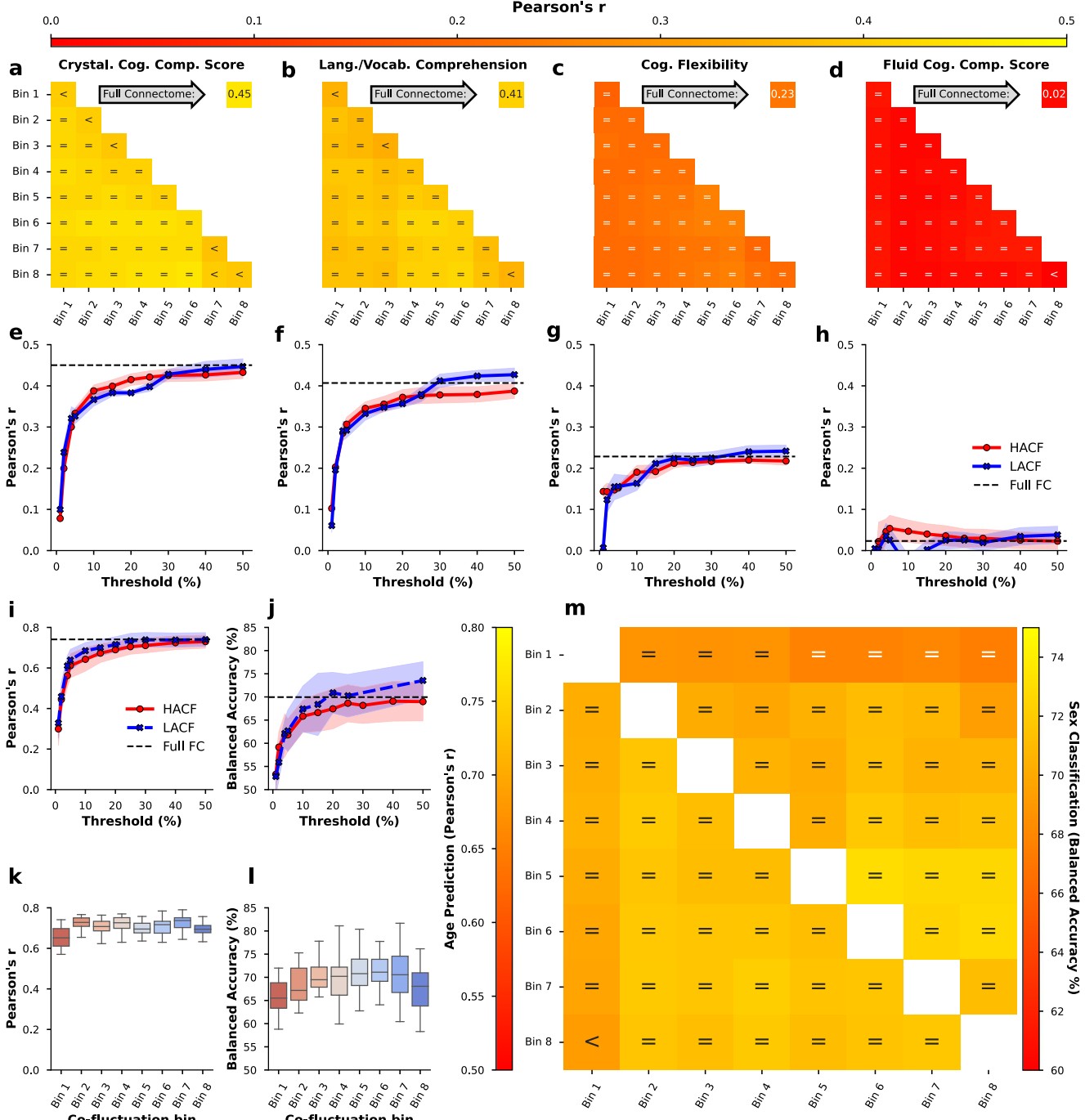

**Fig. 6 Prediction accuracy (Pearson's r) for four cognitive targets as well as age and sex prediction scores in the HCP-A dataset.** The figure displays results for prediction of cognitive targets using the individual and combined bins strategy (**a–d**; individual bins are on the diagonal) and the sequential sampling strategy (**e–h**). Upper and lower boundary of the fill colours indicate the standard deviation across repeats. In addition, results are shown for age prediction (Pearson's r) and sex classification (balanced accuracy) using the sequential (**i**, **j**), individual bins (**k**, **l**), and combined bins (**m**) sampling strategies. Threshold refers to the percentage of highest (HACF) or lowest (LACF) co-fluctuation time points chosen to estimate FC in the sequential sampling strategy and the y-axis displays the prediction scores. In (**k**, **l**) the box plot indicates the median (centre line) and the interquartile range. In (**m**) comparison operators indicate whether scores obtained by a co-fluctuation bin are equivalent to scores obtained by full FC (=) or whether they are less (<) or greater (>) than scores obtained by full FC according to a 5% Bayesian ROPE[43]. Source data can be obtained from Supplementary Data 2 for plots **a–h**, and Supplementary Data 3 for plots **i–m**.

suggest that these weakly synchronised co-fluctuation amplitudes are less stable within a subject but even more variable across subjects.

Next, we systematically examined the predictiveness of different levels of co-fluctuation using machine learning based prediction analyses. In Zamani Esfahlani et al.[18], an exploratory

brain-behaviour correlation analysis was performed. The authors performed PCA on 158 behavioural, trait, and demographic variables to obtain a first principal component (PC1) explaining 20.3% of behavioural variance. Correlations with top 5% HACF time points as well as bottom 5% LACF time points were weak, but relatively stronger for top 5% HACF time points. In our

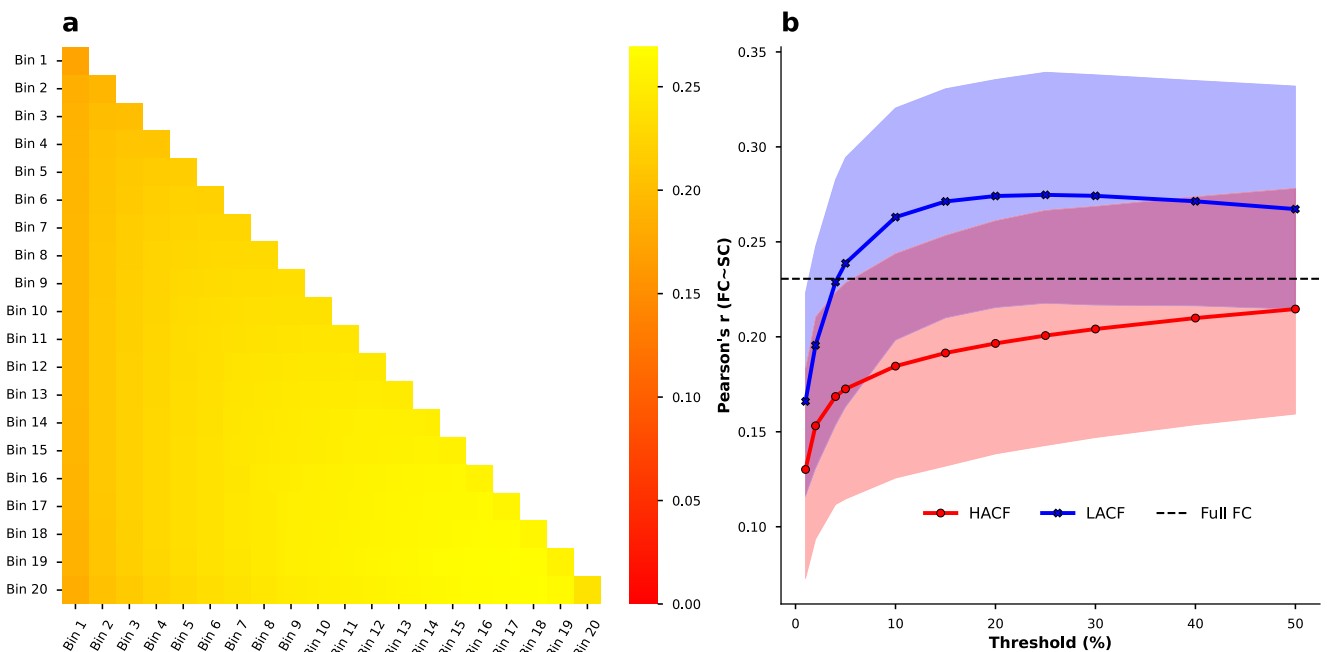

**Fig. 7 Correlations between SC and FC averaged across subjects in the HCP-YA sample.** Results are shown for (**a**) individual (on-diagonal) and combined bins (off-diagonal) and the (**b**) sequential paradigm. In (**b**) upper and lower bounds of fill colours indicate minimum and maximum correlations across subjects. Threshold (x-axis) refers to the percentage of highest (HACF) or lowest (LACF) co-fluctuation time points chosen to estimate FC and the y-axis shows the Pearson correlation between FC edges and SC edges. Source data can be obtained from Supplementary Data 4.

study, we also found that the predictive capacity of FC estimates can indeed be improved by leveraging information from the ETS to select a specific set of points from the original BOLD time series. In contrast to Zamani Esfahlani et al.[18], we found lower predictiveness with HACF time points and higher predictiveness with LACF and especially intermediate time points. This suggests that time points of low to intermediate levels of co-fluctuation may be best suited to make meaningful predictions. Using this insight in preprocessing pipelines may ultimately improve applicability of FC as an imaging-based biomarker of brain function in precision medicine and psychiatry.

A pivotal question associated with our findings concerns the origin of HACF and LACF events. Non-significant correlations have been shown between co-fluctuation amplitudes and breathing data, heart rate and motion (see Figs. S1 and S2 in ref.[18]), indicating that HACF events are unlikely to be driven by physiological noise only. Two main possibilities remain consistent with our results: On one hand there is evidence that HACF events are driven by tasks or change of cognitive state as demonstrated by the finding that RSS time series are more synchronised across participants during movie-watching than during resting state[18]. On the other hand it has been suggested that high RSS amplitudes are simply the result of extreme values in a noisy, but stationary distribution during rs-fMRI[50,51]. Temporal spacing of time points is likely an important factor to consider due to (temporal) autocorrelation[50]. Briefly, the reason that a few time points may suffice to faithfully reconstruct full FC, may be due to the fact that selected time points are well spaced along the time series, and therefore capture a broad range of points from that distribution. This may also play a relevant role in determining the predictive utility of specific FC estimates.

Assuming the validity of the first hypothesis, LACF and intermediate bins may then contain time points of brain activity during which estimated FC corresponds to a greater degree to a baseline brain state closer to underlying structural connectivity. To test this hypothesis, we investigated the correlation between FC at different levels of co-fluctuation and structural connectivity, measured by probabilistic tractography. Our findings indeed show higher correlations between FC during LACF and intermediate time points and structural connectivity (SC), just as another study showing that coupling between estimates of SC and FC was stronger during time points of intermediate or low levels of co-fluctuation[52]. This suggests that time resolved FC may inform us about how structural constraints drive functional organisation of the brain. In addition, language-related tasks (HCP Young adult: reading, vocabulary; HCP Aging: crystallized intelligence, language/vocabulary comprehension) were more predictable using intermediate and LACF time points in contrast to HACF. This observation supports the hypothesis of a stronger link between SC and LACF time points. It is also in line with the previous studies showing strong mapping between language and anatomy[53]. However, the predictive capacity of LACF and intermediate time points did not decrease after regressing out structural connectivity, suggesting these time points capture individual-level phenotypic information independent of brain structure (Fig. 8). It seems unlikely therefore that the effect can only be explained by higher similarity with structural connectivity.

In light of these findings, it would be informative to investigate the properties of ETS in task-based fMRI (t-fMRI). If HACF events are in fact influenced by external stimuli, then one would expect more frequent HACF events after null model thresholding during task engagement. Otherwise, if the co-fluctuation levels and FC estimates over time do not reject stationary null models, then one could conclude that HACF events are simply the result of a random stationary process. Designing relevant null distributions for HACF/LACF of t-fMRI is an avenue for future work. A non-trivial challenge is due to the fact that computation of ETS requires z-scoring of the ROI time series, which is only appropriate if sample mean and standard deviation are time invariant[18,54]. A potential solution could be to regress out the block design from t-fMRI and use the residuals for this type of analysis.

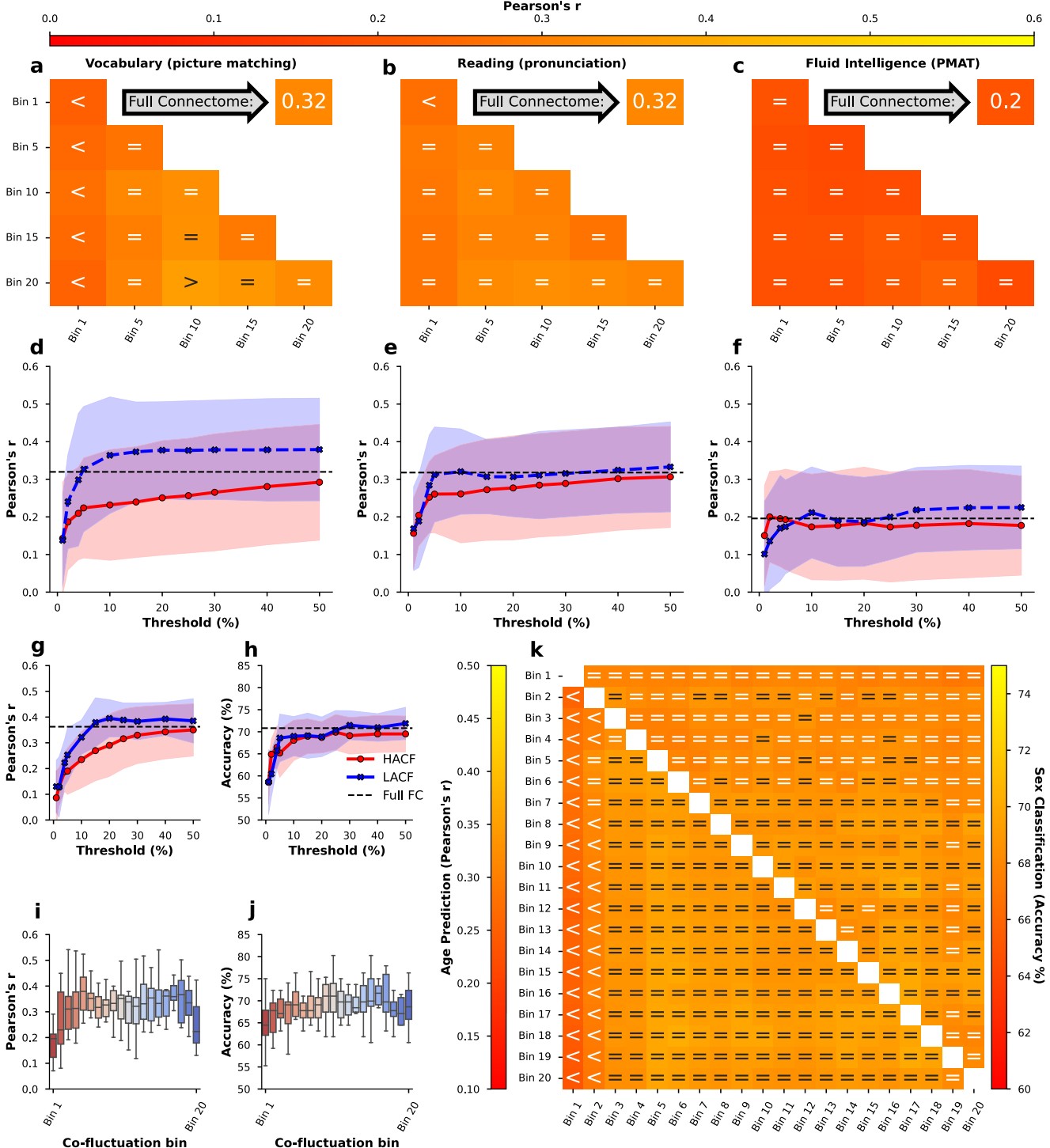

**Fig. 8 Prediction accuracy for three cognitive targets in the HCP-YA sample (Pearson's r between observed and predicted targets) as well as age and sex prediction when removing SC from FC estimates for each subject.** Prediction of the cognitive targets in the combined and individual (**a**–**c**) and the sequential (**d**–**f**) sampling strategies as well as age prediction (Pearson's r) and sex classification (accuracy) using the sequential (**g**, **h**), individual bins (**i**, **j**), and combined bins (**k**) sampling strategies after removal of SC from FC. Threshold refers to the percentage of highest (HACF) or lowest (LACF) co-fluctuation time points chosen to estimate FC in the sequential sampling strategy. In (**i**, **j**) the box plot indicates the median (centre line) and the interquartile range. In (**k**) comparison operators indicate whether scores obtained by a co-fluctuation bin are equivalent to scores obtained by full FC (=) or whether they are less (<) or greater (>) than scores obtained by full FC according to a 5% Bayesian ROPE[43]. Source data can be obtained from Supplementary Data 5.

**Limitations**. One possible limitation when predicting behavioural variables is the unknown influence of confounding factors. In particular, the influence of in-scanner motion has been found to confound brain-behaviour relationships[55]. The impact of motion may also become influential in the identification and identifiability analysis through adding a highly personalised signature to the fMRI datasets[56]. However, as pointed out above, correlations between co-fluctuation amplitudes and physiological noise and motion have been found to be non-significant[18]. Further, the preprocessing strategies employed here to remove nuisance variables and influences of motion from the neural signal have been found to be among the most effective in the literature[57–59]. In addition, we also removed framewise displacement (FD; a measure of in-scanner head motion) from the predicted targets using a linear model. This was done in a CV-consistent fashion, meaning that a linear model was trained separately on the training data for each train-test split in the CV to avoid any test-to-train leakage of information regarding labels in the test set[60]. Lastly, we checked the correlations between RSS and FD for every subject and session, and found that correlations followed a normal distribution centered at zero (see Supplementary Fig. S20).

Moreover, there may be other machine learning models not applied here, which perform better on FC estimates obtained during HACF time points than on FC estimates obtained during intermediate or LACF bins. However, in order to minimise this risk, we applied CBPM[4,9] and kernel ridge regression, two models which are commonly used for FC-based prediction, and have been consistently found to yield competitive results in the prediction of cognitive and demographic variables[4,12,22,44,61]. Both models show results consistent with our conclusions here. Furthermore, we used three distinct models (linear SVM, RBF SVM and a ridge classifier) for sex classification, each of which confirmed the overall pattern that intermediate and LACF bins yield better prediction results than HACF bins.

**Conclusions and future research**. It has previously been suggested that HACF time points capture more individual-level information[18]. However, our findings suggest that time points with intermediate levels of co-fluctuation yield highest subject specificity and predictive capacity of individual-level phenotypes compared to HACF. We further find that assessments of subject specificity provide more robust conclusions when multiple metrics are used (i.e. $I_{diff}$ and $I_{Acc}$). Overall, our findings suggest that intermediate time points may be more informative in individual-level inference and they may inform future preprocessing strategies aiming at identifying robust brain-based biomarkers.

## Methods

**Datasets - Human Connectome Project (HCP)**. The details regarding collection of behavioural data, fMRI acquisition, and image preprocessing in the HCP Young-Adult (HCP-YA) project[37,38,62] as well as the HCP Aging (HCP-A) project[39,40] have been described elsewhere. Here, we aim to give a quick overview. The scanning protocol for both HCP-YA and HCP-A was approved by the local Institutional Review Board at Washington University in St. Louis. Retrospective analysis of these datasets was further approved by the local Ethics Committee at the Faculty of Medicine at Heinrich-Heine-University in Düsseldorf.

*HCP young-adult (HCP-YA)*. We used data obtained from four resting-state fMRI (rs-fMRI) sessions taken from the HCP-YA S1200 release[38]. Subjects were selected if data was available for all four resting state sessions and 25 predefined behavioural variables of interest. This resulted in a dataset consisting of 771 subjects (384 female, 387 male). Participants' age ranged from 22 to 37 ($M = 28.41$, $SD = 3.74$). The four sessions of rs-fMRI were obtained on two separate days (each lasted ca. 15 min; ~60 minutes across all four sessions). On each day, two sessions were recorded for different phase encoding directions (left-right [LR] and right-left [RL]) providing four overall rs-fMRI datasets. Scans were acquired using a 3T Siemens connectome-Skyra scanner with a gradient-echo EPI sequence (TE =

33.1 ms, TR = 720 ms, flip angle = 52°, 2.0 mm isotropic voxels, 72 slices, multiband factor of 8).

*HCP aging (HCP-A)*. We used a subset of the HCP-A dataset as validation data to replicate our main findings from the exploratory HCP-YA dataset. Similar to the HCP-YA, in the HCP-A two sessions of rs-fMRI were acquired on two separate days with a 2D multiband (MB) gradient-recalled echo (GRE) echo-planar imaging (EPI) sequence (MB8, TR/TE = 800/37 ms, flip angle = 52°) and 2.0 mm isotropic voxels covering the whole brain (72 oblique-axial slices) using a Siemens 3T Prisma scanner. For each session, functional scans were acquired in two separate runs with opposite phase encoding polarity (anterior-to-posterior [AP] and posterior-to-anterior [PA]). Subjects who did not have data for all four of these runs were excluded. Further we only included subjects that had data for all four selected behavioural targets and confounding variables. This resulted in a sample of 558 subjects (316 female, 242 male) with ages ranging between 36 and 100 years ($M = 59.87$, $SD = 15.03$).

Data from rs-fMRI sessions in both of these datasets (HCP-YA and HCP-A) had also already undergone the HCP's minimal preprocessing pipeline[37], including motion correction and registration to standard space. In addition, the ICA-FIX procedure (independent component analysis and FMRIB's ICA-based X-noiseifier[58]) was applied to remove structured artefacts. Lastly, the 6 rigid-body parameters, their temporal derivatives and the squares of the 12 previous terms were regressed out, resulting in 24 parameters. Any further confound removal and preprocessing was applied to this denoised data.

**Image pre-processing**. For both datasets, we regressed out confounds, linearly detrended and bandpass filtered the signal at 0.008–0.08 Hz using "nilearn.image.clean_img". For the main analysis, this included mean time courses of the white matter (WM), cerebro-spinal fluid (CSF), and global signal (GS), as well as their squared terms, and the temporal derivatives of the mean signals as well as their squared terms as confounds, resulting in 12 parameters (4 for each noise component). In the supplementary we also provide results without global signal regression. A binary spike regressor was further added for each fMRI frame exceeding a motion threshold (i.e. 1 where root mean squared framewise displacement [FD] > 0.25 mm, and 0 where FD < 0.25 mm). The resulting voxel-wise images were then aggregated into the Schaefer 200 parcellation[41]. In the supplementary information, we also provide results using the Schaefer 300 and 400 parcellation, and without the use of global signal regression. In addition, in the HCP-A dataset time series were cut by excluding the first 20 and the last 18 volumes of the scan, so that the resulting time series consisted of 440 volumes that could be divided into 8 bins of 55 volumes. This was done to ensure that bins were of comparable size in both datasets, since the time series in the HCP-YA dataset (1200 volumes each) were divided into 20 bins of 60 volumes each.

**Edge time series construction and functional connectivity estimation**. Edge time series were computed as described in ref. [18]. The parcellated BOLD time series were z-scored. Then, the element-wise product between the z-scored timeseries of each pair of parcels was computed as an estimate of co-fluctuation between parcels over time. The magnitude of co-fluctuation was quantified using the root sum of squares at each time point (RSS) resulting in a co-fluctuation time series for each subject. Afterwards, the time points in the BOLD time series were ordered according to their corresponding RSS (from high to low) for every subject.

To test whether HACF moments capture more meaningful information about individual subjects than LACF moments, we sampled time points using three separate strategies. For every subject and every resting-state session in the HCP dataset, the BOLD time series was ordered according to co-fluctuation magnitude. Each strategy differs in selection of time points used to construct the FC. In strategy (1) (individual bins sampling) the ranked BOLD time series was divided into twenty bins each containing 5% of the time series (60 time points) in the case of the HCP-YA dataset. From these twenty bins, five bins were sampled to be used in prediction. We did not consider all twenty bins in the HCP-YA in the prediction analysis, because running the pipeline for every bin and each of the targets would incur unnecessarily high and impractical computational cost. For each of the selected bins, FC was estimated using pairwise Pearson's correlation coefficients. In the case of the HCP-A, the ranked BOLD time series was divided into 8 bins, each containing 12.5% of the timeseries (55 time points). All 8 bins were used in prediction. In strategy (2) (combined bins sampling), we sampled every possible combination of two bins out of the defined individual bins used in strategy (1) and concatenated these bins to estimate FC. In strategy (3) (sequential sampling), we used HACF and LACF time points, but applied sequentially increasing thresholds to include varying numbers of time points on either side. In each sampling strategy, FC estimates of corresponding co-fluctuation bins were first averaged across the phase encoding directions, resulting in two FC matrices per subject per co-fluctuation bin, to be used in identification. These two FC matrices were further averaged resulting in one FC matrix per subject per co-fluctuation bin to be used in prediction.

Each sampling strategy was chosen with a specific goal in mind: The first strategy, which we referred to as the sequential strategy, was used to replicate and extend the findings of a previous study[18]. The second strategy, which we called the individual bins strategy, was designed to investigate the intermediate bins that were

not examined in the previous study[18]. The third strategy that we employed in this study was to combine individual bins to test whether combining bins from very different co-fluctuation levels also provides additional information, or whether behavioural information is maximised by any one level of co-fluctuation. This strategy was selected to examine the potential for shared information between co-fluctuation levels.

**Structural connectivity extraction**. Diffusion-weighted magnetic resonance imaging (dMRI) data had already been processed using the HCP diffusion minimal preprocessing pipeline[37]. This included normalisation of the b0 image intensity across runs as well as removing EPI distortions, eddy-current-induced distortions and subject motion. It further corrected for gradient-nonlinearities and diffusion data was registered to the structural T1w scan. Structural connectivity (SC) matrices were extracted from these preprocessed dMRI data using a workflow developed in-house[63]. Ten million total streamlines of the whole-brain probabilistic tractography (WBT) were calculated using MRtrix3. Response functions were estimated using a three-tissue constrained spherical deconvolution algorithm[64]. Fibre oriented distributions (FOD) were estimated from the dMRI data using spherical deconvolution, and the WBT was created through the fibre tracking by the second-order integration over the FOD by a probabilistic algorithm[65]. The tracking parameters were set as default values for the tckgen function from MRtrix documentation (https://mrtrix.readthedocs.io), where the following values were used: step size = 0.625 mm, angle = 45 degrees, minimal length = 2.5 mm, maximal length = 250 mm, FOD amplitude for terminating tract = 0.06, maximum attempts per seed = 1000, maximum number of sampling trials = 1000, and downsampling = 3. In the atlas transformation, labels were annotated using a classifier to parcel cortical regions in the native T1w space using Freesurfer[66] according to the Schaefer atlas with 200-area parcellation[41]. The pipeline then transformed the labelled image from the T1w to dMRI native space. After the transformation, the labelled voxels in the grey matter mask were selected for a seed and a target region. Consequently, the tck2connectome function of MRtrix3 reconstructed SC. From the original sample of 771 subjects in the identification and prediction analyses, 3 subjects had to be excluded because they lacked dMRI data. Another 6 subjects were excluded due to software errors during structural preprocessing, resulting in a sample of 762 subjects (383 female, 379 male) with ages ranging from 22 to 37 ($M = 28.41$, SD = 3.73) for all analyses involving structural connectivity.

**Subject specificity: differential identifiability and identification accuracy**. To assess identifiability of FC for subjects across different sessions, we used both identification accuracy ($I_{acc}$)[4] and the differential identifiability quality function $I_{diff}$[5]. Identification refers to the paradigm by which an individual's FC profile obtained in an fMRI scanning session is used to identify them from a database of FC profiles obtained in a second fMRI scanning session[4]. While identification accuracy is defined as the proportion of correctly identified participants, differential identifiability is defined as the difference between mean within-subject correlations ($I_{self}$) and mean between-subject correlations ($I_{other}$): $I_{diff} = (I_{self} - I_{other}) * 100$[5,7]. Higher levels of differential identifiability indicate a stronger individual fingerprint.

**Prediction of behavioural and demographic measures**. Prediction was performed with FC matrices obtained using the three distinct strategies outlined above and averaged across all four HCP resting state sessions with unique edges serving as features. We selected phenotypes used in ref. [42] from the categories Cognition, In-scanner task performance, and Personality (see Table S1) as targets, resulting in 25 targets overall. In the HCP-A dataset we used four cognitive targets. We selected Language/Vocabulary Comprehension and Cognitive Flexibility since these measures were also available in the HCP-YA sample and showed reasonable prediction accuracy. As there was no direct test of fluid intelligence in the HCP-A sample, we also included composite measures of fluid and crystallised cognition (see Table S2). Detailed descriptions of these targets can be found elsewhere[40,62].

To control for confounding influences, age at scan, sex at birth and framewise displacement (FD) of resting state fMRI recordings were regressed out from the targets in a CV-consistent fashion as these have been found to correlate with behavioural variables[12,55]. In this context, CV-consistency means training the confound regression model on the training partition of a split only, and applying this model to the test data subsequently, to avoid test-to-train leakage. In addition to prediction of behavioural targets we also predicted age and sex. In the prediction of age, only sex and FD were removed as confounds. In the prediction of sex, we removed age, brain volume (FS_BrainSeg_Vol), educational status (SSAGA_Educ), and FD as confounds. In the HCP-YA dataset, one subject (male) had to be excluded from sex prediction due to missing information on confounds (SSAGA_Educ).

For all regression tasks, we used ridge regression with a Pearson kernel. This model has been recommended as an efficient way to benchmark predictive utility of FC representations, and it performs well even compared to sophisticated deep learning algorithms specifically designed for connectivity-based features[12,21,22]. We further validated our main findings using CBPM[9]. In sex classification, we used a ridge classifier as well as support vector classifiers (SVC) with a linear kernel or a radial basis function (RBF) kernel, to see whether results are robust across different models.

To assess out-of-sample prediction accuracy in the HCP-YA dataset, a 10-fold nested cross-validation (CV) was performed for each FC representation. The folds were split such that family members were always within the same fold, so that independence between folds was maintained. To select the l2-regularisation strength for ridge regression and classification as well as the C parameter for the support vector classifiers in CV-consistent fashion, we used a 5-fold inner CV on the training folds. Candidate values for the l2-regularisation strength used in hyperparameter tuning were: {0, 0.00001, 0.0001, 0.001, 0.004, 0.007, 0.01, 0.04, 0.07, 0.1, 0.4, 0.7, 1, 1.5, 2, 2.5, 3, 3.5, 4, 5, 10, 15, 20, 30, 40, 50, 60, 70, 80, 100, 150, 200, 300, 500, 700, 1000, 10,000, 100,000, 1,000,000}. Candidate values for the C parameter of the support vector classifiers were 50 numbers between 0.01 and 100 evenly spaced on the log scale generated by the "numpy.geomspace" function.

The model with the best parameters was fitted on the training folds and then tested on the outer CV test fold. In the HCP-A dataset we used a 5-fold nested CV with five repetitions, since we only included unrelated subjects, and therefore had no grouping constraint. In addition, the number of samples was lower, so a 5-fold CV could ensure that test folds have sufficient samples. To evaluate prediction accuracy, we report Pearson's r and the coefficient of determination ($R^2$) for regression tasks as well as the mean absolute error (MAE) in the case of age prediction. For predicted values $\hat{y}$ and corresponding observed values $y$ over $n$ samples with a sample mean of the observed values $\bar{y}$, these metrics are defined as:

**Pearson's r**

$$r_{y\hat{y}} = \frac{n\sum_{i=1}^{n} y_i\hat{y}_i - \sum_{i=1}^{n} y_i \sum_{i=1}^{n} \hat{y}_i}{\sqrt{n\sum_{i=1}^{n} y_i^2 - (\sum_{i=1}^{n} y_i)^2}\sqrt{n\sum_{i=1}^{n} \hat{y}_i^2 - (\sum_{i=1}^{n} \hat{y}_i)^2}} \quad (1)$$

**Coefficient of determination**

$$R^2(y, \hat{y}) = 1 - \frac{\sum_{i=1}^{n}(y_i - \hat{y}_i)^2}{\sum_{i=1}^{n}(y_i - \bar{y})^2} \quad (2)$$

**Mean absolute error (MAE)**

$$MAE(y, \hat{y}) = \frac{1}{n}\sum_{i=1}^{n}|y_i - \hat{y}_i|. \quad (3)$$

Unlike Pearson's r and $R^2$, the MAE is not scale invariant and therefore more difficult to interpret when predicting psychometric variables on different scales. In sex classification we used accuracy in the HCP-YA dataset:

**Accuracy**

$$Acc(y, \hat{y}) = \frac{1}{n}\sum_{i=1}^{n} 1(\hat{y}_i = y_i). \quad (4)$$

Due to imbalanced distribution of the class labels (316 female, 242 male), balanced accuracy was used in the HCP-A dataset to avoid inflated performance estimates:

**Balanced accuracy**

$$Acc_{balanced}(y, \hat{y}) = \frac{1}{2}\left(\frac{TP}{TP + FN} + \frac{TN}{TN + FP}\right) \quad (5)$$

where $TP$, $FP$, $TN$, and $FN$ denote true positives, false positives, true negatives, and false negatives respectively[46].

**Bayesian region-of-practical-equivalence (ROPE) approach**. To compare prediction accuracy between individual or combined bins and the full FC, we used a Bayesian 'region of practical equivalence' (ROPE) approach[43]. In this approach one defines two models as practically equivalent if differences between accuracy scores do not exceed a pre-defined percentage. It is a statistical approach used in Bayesian inference to determine a region around a null value (our predefined percentage) in which the posterior probabilities of a given parameter falling within this region can be determined using Bayesian estimation[67]. Specifically, the Bayesian estimation starts with a prior distribution. Typically, a reasonable assumption as a prior distribution would be that the algorithms perform equally well. Here, we adopt as prior the normal distribution as suggested in ref. [43]. Using the prior and the results of the experiment (the observed data), a posterior distribution can then be used estimated. We can then estimate three different probabilities regarding differences in model accuracies (i.e. in relation to our hypotheses). Assuming that differences $(x - y)$ are obtained between two models $x$ and $y$:

1. $P(x < y)$: the posterior probability that model $y$ performs better than $x$; this is the integral of the distribution to the left of the region of practical equivalence (i.e. where differences are negative)
2. $P(x = y)$: the posterior probability that model $x$ and $y$ are practically equivalent; this is the integral of the distribution inside the region of practical equivalence
3. $P(x > y)$: the posterior probability that model $x$ performs better than $y$; this is the integral to the right of the region of practical equivalence (i.e. where differences are positive)

That is, the posterior probabilities represent the degree of belief in a given hypothesis after taking into account the observed data. One advantage of Bayesian estimation is that it does not rely on a point-wise null hypothesis but rather on a

range of potential values within the ROPE. This ROPE can be specified to reflect the practical context. For example, if even a small difference in accuracy can incur cost that is considered practically meaningful within a given context one may go for a ROPE of 1% (i.e. any two models that differ by less than 1% would be considered practically different). In our study, we used a 5% ROPE, so that even 5% differences in model accuracy would be considered practically equivalent.

**Statistics and reproducibility**. In the HCP-YA dataset, the identification analyses were conducted using a sample of 771 subjects (384 female, 387 male). Participants' age ranged from 22 to 37 ($M = 28.41$, SD = 3.74). In the HCP-A dataset, the identification analyses were conducted using a sample of 558 subjects (316 female, 242 male) with ages ranging between 36 and 100 years ($M = 59.87$, SD = 15.03). In both datasets, identification was performed between rs-fMRI data from day 1 and day 2 of data collection and for each day FC was averaged across phase encoding directions. Identification was performed with day 1 data as a source and day 2 data as a target, and vice versa resulting in two identification accuracy scores per co-fluctuation bin. These two scores were averaged to obtain one overall identification accuracy. Differential identifiability was also estimated between rs-fMRI data from day 1 and day 2. In order to obtain an estimate of the variance, and to assess whether differences in identification accuracy and differential identifiability are statistically significant, we performed a bootstrapping procedure via resampling the original samples with replacement 1000 times. In each resampling run only unique subjects from the resampled subject list were chosen to perform the identification experiment, resulting in 1000 identification accuracy scores as well as 1000 differential identifiability scores per co-fluctuation bin. For each co-fluctuation bin, scores for HACF- and LACF-derived FC were compared with two-tailed Wilcoxon-signed-rank tests ($n = 1000$), resulting in one p-value per co-fluctuation bin. To account for multiple comparisons, a Bonferroni correction at the alpha level of significance of 0.05 was applied.

In both datasets, the exact same samples were also used for machine-learning-based prediction analyses. In the HCP-YA, for every co-fluctuation bin, a grouped 10-fold cross-validation was used to estimate the generalisation error of each model to account for the dependent family structure of this dataset. In the HCP-A dataset, only unrelated subjects were included in the sample. In addition, the HCP-A sample was slightly smaller, so we used a 5-fold cross-validation with five repetitions to make sure that each fold had sufficient samples. In the combined bins sampling strategy, we used a Bayesian ROPE approach to compare each model train on a given co-fluctuation bin to a model trained on all of the data available to estimate whether specific subsets of all time points can meaningfully improve predictive capacity.

Analyses conducted on structural connectivity were only performed for the HCP-YA dataset. From the original sample of 771 subjects in the identification and prediction analyses, three subjects had to be excluded because they lacked dMRI data. Another six subjects were excluded due to software errors during structural preprocessing, resulting in a sample of 762 subjects (383 female, 379 male) with ages ranging from 22 to 37 ($M = 28.41$, SD = 3.73) for all analyses involving structural connectivity. Similarity between structural connectivity and functional connectivity was assessed using Pearson's correlation coefficient.

**Reporting summary**. Further information on research design is available in the Nature Portfolio Reporting Summary linked to this article.

## Data availability

Further information on how to obtain the HCP-YA and the HCP-A datasets can be obtained at https://www.humanconnectome.org/. Data and/or research tools used in the preparation of this manuscript were obtained from the National Institute of Mental Health (NIMH) Data Archive (NDA). NDA is a collaborative informatics system created by the National Institutes of Health to provide a national resource to support and accelerate research in mental health. Dataset identifier: [https://doi.org/10.15154/1527952]. This manuscript reflects the views of the authors and may not reflect the opinions or views of the NIH or of the Submitters submitting original data to NDA. Source data for the plots in the figures are available in supplementary data files. Moreover, we provide a supplementary information file containing results for additional robustness checks and analyses. Any remaining information can be obtained from the corresponding author upon reasonable request.

## Code availability

Code used to generate edge time series, connectomes, and perform prediction and other analyses on these data can be found in a public GitHub repository (https://github.com/juaml/etspredict). The code used to obtain SC is available at https://jugit.fz-juelich.de/inm7/public/vbc-mri-pipeline.

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

## Acknowledgements
This work was supported by the Helmholtz Portfolio Theme "Supercomputing and Modelling for the Human Brain" and the European Union's Horizon 2020 Research and Innovation Programme under Grant Agreement No. 945539 (HBP SGA3), the DFG Priority Program (SPP2041) project number 454012190, El 816 28-1 (Machine-learning on Brain Connectomics: Individual Prediction of Cognitive Functioning in Health and Cerebreal Small Vessel Disease), and by the Max Planck School of Cognition supported by the Federal Ministry of Education and Research (BMBF) and the Max Planck Society (MPG).

## Author contributions
K.R.P. developed the idea of the study. K.R.P., L.S., D.I.L., and A.O. conceptualised the study. L.S. preprocessed functional MRI data and performed machine learning and identification analyses. L.S., K.J., and F.H. preprocessed the structural connectivity data. L.S. prepared the results, including figures and tables. L.S. and D.I.L. drafted the manuscript together. All authors including S.B.E and G.J. commented and contributed to the final manuscript. This work has been done in partial fulfilment of the requirements for a PhD thesis.

## Funding

## Competing interests
The authors declare no competing interests.
