## [Peer Review File · Communications Biology]

Reviewers' comments:

Reviewer #1 (Remarks to the Author):

The authors have focused on the contribution of different edge time series time points on the relation between the brain and behavior. Via machine learning approaches, the authors tried to investigate the functional connectivity (FC) capability in differential identifiability and identification accuracy. The results showed that lower and intermediate parts of ETS have the highest effect on both subject identifiability and predictiveness.

The work is a potentially important one in the field of finding the relationship between functional connectivity and behavior. The authors investigated this association from different points of view by considering three different strategies. In each strategy, they select different time frames with different levels of co-fluctuation. The authors need to address the questions/criticisms (see below) properly and make further investigations to deserve a publication in Communications Biology.

Major Comments:

1. This study lacks statistical analysis. The authors report several interesting results without statistical tests. For instance, it is mentioned that "In the sequential sampling strategy, we could replicate the finding that HACF frames yield higher differential identifiability (IDiff) than LACF frames (Fig. 1A)." However, there is no statistical report on this difference to clarify if the difference between HACF and LACF is significant or not. As another example in figure 2, there is no detailed information about $= < >$ signs, there are no statistical reports about values such as correlation and p-values.

2. The aim of defining three different selection strategies is not clear in the manuscript. In other words, the authors did not mention which information they can get from each strategy.

3. The authors defined three strategies. For two of them "individual" and "combined" three states, low, intermediate, and high amplitude co-fluctuation are considered. While for the third one, sequential, the authors just considered low and high amplitude co-fluctuation. While the major finding of this work is related to intermediate bins.

Minor Comments:

1. The HCP dataset has 4 different sessions, I would suggest the authors clarify which sessions were used for this study.

2. Captions of some figures, such as figures 1 and 5 need to be completed. The captions are not sufficiently provided information about how the figures were obtained and what they illustrated.

Suggestion: Before resubmitting, I would suggest the authors carefully go through the article and break the long sentences into short and clear sentences.

Reviewer #2 (Remarks to the Author):

Sasse and colleagues study temporal fluctuations in FC within the duration of a functional magnetic resonance

imaging (fMRI) scanning session. Employing three time frame sampling strategies, we investigated predictiveness of targets across the domains of cognition, behaviour, personality, 66 and demographics from FC including all frames. Given the widespread use of functional connectivity measures as features for predictive models in neuroimaging, this paper can make a valuable contribution to the research literature. This reviewer recommends integration of the following action

points:

Major

1) Introduction: first paragraph appears quite general and could get faster at the research question at hand

2) Introduction / "These time points of high-amplitude co-fluctuations, called "events", contribute disproportionately to FC and are thought to reflect fluctuations in cognitive state": not all high-magnitude change in resting state signals are thought to be cognitively or behaviourally relevant; please smoothen statement by integrating:

Uddin, L. Q. (2020). Bring the noise: reconceptualizing spontaneous neural activity. *Trends in Cognitive Sciences*, 24(9), 734-746.

Li, J., Bolt, T., Bzdok, D., Nomi, J. S., Yeo, B. T., Spreng, R. N., & Uddin, L. Q. (2019). Topography and behavioral relevance of the global signal in the human brain. *Scientific reports*, 9(1), 1-10.

Bzdok, D., Varoquaux, G., Grisel, O., Eickenberg, M., Poupon, C., & Thirion, B. (2016). Formal models of the network co-occurrence underlying mental operations. *PLoS computational biology*, 12(6), e1004994.

3) Methods: why and how were the 2 resting-state (out of 4) from the HCP data selected? Perhaps its is wording, different phase encoding acquisitions could be different sessions, too.

4) Methods: more information should be provided on whether and how the global signal regression was present and computed in the main analysis?

5) Methods: more information could/should be provided on how the spike regression for signal cleaning was carried out quantitatively.

6) Methods / connectome prediction: where the features cleaned or encoded different? For what reasons where the kernels picked in ridge regression modeling? Was any hyper parameter selection done, and, if yes, how? How was the confounding removal integrated into the cross validation workflow?

Poldrack, R. A., Huckins, G., & Varoquaux, G. (2020). Establishment of best practices for evidence for prediction: a review. *JAMA psychiatry*, 77(5), 534-540.

7) General: the manuscript appears to be meta-stable between 'identification' and 'identifiability', which are similar, but different notions - please keep and argue for one of them throughout the whole work.

Minor

1) Better to avoid abbreviations in subtitles.

2) figures: the 'threshold' axis may be hard to follow for several readers and could be explained more. Also the nature of the metric on the y axis is not always directly apparent.

**Author rebuttal to reviews of “Intermediately Synchronised**
**Brain States optimise Trade-off between Subject Identifiability**
**and Predictive Capacity”**

**Referee #1**

*The authors have focused on the contribution of different edge time series time points on the*
*relation between the brain and behavior. Via machine learning approaches, the authors tried to*
*investigate the functional connectivity (FC) capability in differential identifiability and identification*
*accuracy. The results showed that lower and intermediate parts of ETS have the highest effect on*
*both subject identifiability and predictiveness. The work is a potentially important one in the field of*
*finding the relationship between functional connectivity and behavior. The authors investigated this*
*association from different points of view by considering three different strategies. In each strategy,*
*they select different time frames with different levels of co-fluctuation. The authors need to address*
*the questions/criticisms (see below) properly and make further investigations to deserve a*
*publication in Communications Biology.*

**Major Comments:**

**1. Major Comment:**

*This study lacks statistical analysis. The authors report several interesting results without statistical*
*tests. For instance, it is mentioned that “In the sequential sampling strategy, we could replicate the*
*finding that HACF frames yield higher differential identifiability (IDiff) than LACF frames (Fig. 1A).”*
*However, there is no statistical report on this difference to clarify if the difference between HACF*
*and LACF is significant or not. As another example in figure 2, there is no detailed information*
*about = < > signs, there are no statistical reports about values such as correlation and p-values.*

**Response and Changes:**

We thank the reviewer for their thorough review of our study. We appreciate the feedback and
understand the concern regarding the statistical analysis of our results. To address this concern,
we distinguish between results obtained in the identification experiments, and the results obtained
in the machine learning experiments. Firstly, we have adjusted and re-run the identification
analysis by leveraging a bootstrapping approach. This allows us to perform meaningful statistical
tests to compare the identification and identifiability metrics. Secondly, we have adjusted the
manuscript to go into more depth regarding the Bayesian “region-of-practical-equivalence” (ROPE)
approach that we use to perform statistical comparisons between the prediction accuracies
obtained using different sampling strategies.

**1. Identification experiments:** Thus far we did not use inferential statistics for the
identification analysis given that each identification problem only results in one accuracy score.
That is, for a given co-fluctuation bin, we perform identification once with the REST1 session as a
source and the REST2 session as a target (REST1-REST2), resulting in one score, and then vice
versa (i.e. using REST2 as a source and REST1 as a target (REST2-REST1)). Similarly,
differential identifiability between two sessions always results in only one score. Therefore, with
only two data points we could not use inferential statistics to compare different co-fluctuation bins.

In order to alleviate this problem, we have now changed and re-run this analysis using a
bootstrapping approach. We have incorporated these changes into the manuscript on page 4, lines
108 to 123 in the “**Results**” section under “**Differential identifiability and identification**
**accuracy disagree in their assessment of functional connectivity “fingerprints”**”:

“... To assess variance in identification accuracy and differential identifiability, we
resampled the original 771 subjects 1000 times with replacement, keeping only the unique subjects
from the resampled subject list. In the sequential sampling strategy, we could replicate the finding
that HACF frames yield higher differential identifiability (I_{Diff}) than LACF frames (Fig. 1a). To assess
statistical significance of this observation we used two-tailed Wilcoxon-signed-rank tests with a
Bonferroni correction for multiple comparisons. Across all specified sampling thresholds, HACF-
derived FC provided statistically significantly higher differential identifiability scores at the .05 alpha
level of significance. At the same time, the sequential sampling strategy shows that LACF frames
provide statistically significantly higher identification accuracy (I_{Acc}) than HACF frames across all
specified sampling thresholds (Fig. 1b). Moreover, it becomes evident across the individual and
combined bins sampling strategies that the highest I_{Diff} is in fact achieved by intermediate bins (Fig.
1c). In the individual and combined bins sampling strategies, bins of intermediate co-fluctuation
achieved the highest I_{Acc} overall (Fig. 1d). In the sequential sampling strategy, it is most apparent
that I_{Acc} and I_{Diff} show opposing effects (Fig. 1a, b). ...”

We have also adjusted the “**Materials and Methods**” section to include information about
the bootstrap procedure applied to the identification analysis. We have added the following
paragraph to “**Subject Specificity: Differential Identifiability and Identification Accuracy**” on
page 23, lines 488 to 496:

“... In order to obtain an estimate of the variance, and to assess whether differences in
identification accuracy and differential identifiability are statistically significant, we performed a
bootstrapping procedure via resampling the original 771 subjects with replacement 1000 times. In
each resampling run only unique subjects from the resampled subject list were chosen to perform
the identification experiment, resulting in 1000 identification accuracy scores as well as 1000

differential identifiability scores per co-fluctuation bin. For each co-fluctuation bin, scores for HACF-
and LACF-derived FC were compared with two-tailed Wilcoxon-signed-rank tests, resulting in one
p-value per co-fluctuation bin. To account for multiple comparisons, a Bonferroni correction at the
alpha level of significance of .05 was applied....”

**2. Prediction experiments:** To compare prediction accuracy between individual or
combined bins and the full FC, we used a Bayesian ‘region of practical equivalence’ (ROPE)
approach (Benavoli et al., 2017). We acknowledge that in our paper we did not dedicate a lot of
text towards explaining this approach and we thank the reviewer for pointing this out. To better
explain the rationale of our statistical tests in the prediction analyses, we added a sub-section
entitled “**Bayesian “region-of-practical-equivalence” (ROPE) approach**” to the “**Materials and**
**Methods**” section, where we expand a bit more on this particular method. The text from pages 25
to 26 (lines 560 to 589) was changed accordingly:

“... To compare prediction accuracy between individual or combined bins and the full FC,
we used a Bayesian ‘region of practical equivalence’ (ROPE) approach³⁸. In this approach one
defines two models as practically equivalent if differences between accuracy scores do not exceed
a pre-defined percentage. It is a statistical approach used in Bayesian inference to determine a
region around a null value (our predefined percentage) in which the posterior probabilities of a
given parameter falling within this region can be determined using Bayesian estimation⁶³. Using the
prior and the results of the experiment (the observed data), a posterior distribution can be
estimated. We can then estimate three different probabilities regarding differences in model
accuracies (i.e. in relation to our hypotheses).

Assuming that differences ($x - y$) are obtained between two models x and y :

- 1. $P(x < y)$: the posterior probability that model y performs better than x ; this is the integral
of the distribution to the left of the region of practical equivalence (i.e. where differences are
negative)
- 2. $P(x = y)$: the posterior probability that model x and y are practically equivalent; this is the
integral of the distribution inside the region of practical equivalence
- 3. $P(x > y)$: the posterior probability that model x performs better than y ; this is the integral to
the right of the region of practical equivalence (i.e. where differences are positive)

That is, the posterior probabilities represent the degree of belief in a given hypothesis after taking
into account the observed data. One advantage of Bayesian estimation is that it does not rely on a
point-wise null hypothesis but rather on a range of potential values within the ROPE. This ROPE
can be specified to reflect the practical context. For example, if even a small difference in accuracy
can incur cost that is considered practically meaningful within a given context one may go for a
ROPE of 1% (i.e. any two models that differ by less than 1% would be considered practically

different). In our study, we used a 5% ROPE, so that 5% differences in model accuracy would be
considered practically equivalent. ...”

**2. Major Comment:**

*The aim of defining three different selection strategies is not clear in the manuscript. In other*
*words, the authors did not mention which information they can get from each strategy.*

**Response:**

We acknowledge that the rationale for selecting these strategies was not clearly stated in the
manuscript, and we appreciate the opportunity to provide more clarity and insight into these.

The first strategy, which we refer to as the sequential strategy, was used to replicate and
extend the findings of a previous study by Zamani Esfahlani et al. (2020). This strategy was
chosen to allow us to obtain results that are directly comparable to theirs and to explore the
generalisability of their findings.

The second strategy, which we call the individual bins strategy, was designed to investigate
the intermediate bins that were not examined in the previous study by (Zamani Esfahlani et al.,
2020). Our goal was to extend their findings and to explore the possibility that intermediate bins
may also be behaviorally relevant. We hypothesised that different levels of co-fluctuation in rs-fMRI
may be differently predictive of different behaviours.

A third strategy that we employed in this study was to combine individual bins to test
whether combining bins from different co-fluctuation levels could provide additional information, or
whether behavioural information is maximised by any one level of co-fluctuation. This strategy was
selected to examine the potential for shared information between co-fluctuation levels.

In summary, we chose the three strategies with specific objectives in mind: to replicate and
extend previous findings, to investigate the intermediate bins, and to examine the relationship
between different levels of co-fluctuation. Overall, the use of these three strategies allowed
comprehensive investigation of different aspects of the relationship between co-fluctuation levels
and behaviour. We have adjusted the **“Materials and Methods”** section to better explain these
objectives.

**Changes:**

Accordingly, we have added a paragraph in the **“Materials and Methods”** section, see page 20
(lines 439 to 446). The revised paragraph is shown below for convenience:

“... Each sampling strategy was chosen with a specific goal in mind: The first strategy,
which we referred to as the sequential strategy, was used to replicate and extend the findings of a
previous study¹⁸. The second strategy, which we called the individual bins strategy, was designed
to investigate the intermediate bins that were not examined in the previous study¹⁸. The third

strategy that we employed in this study was to combine individual bins to test whether combining
bins from very different co-fluctuation levels also provides additional information, or whether
behavioural information is maximised by any one level of co-fluctuation. This strategy was selected
to examine the potential for shared information between co-fluctuation levels. ...”

**3. Major Comment:**

*The authors defined three strategies. For two of them “individual” and “combined” three states, low,*
*intermediate, and high amplitude co-fluctuation are considered. While for the third one, sequential,*
*the authors just considered low and high amplitude co-fluctuation. While the major finding of this*
*work is related to intermediate bins.*

**Response:**

We agree with the reviewer that the sequential strategy lacks information on intermediate bins. We
used the sequential strategy to replicate and extend the findings of the previous study by Zamani
Esfahlani et al. (2020), to assess whether our results are comparable to theirs and to explore the
generalisability of their findings. As this strategy indeed disregards intermediate co-fluctuation
levels we chose to additionally perform the individual bins strategy and the combined bins strategy.

**Minor Comments:**

**1. Minor Comment:**

*The HCP dataset has 4 different sessions, I would suggest the authors clarify which sessions were*
*used for this study.*

**Response:**

We thank the reviewer for pointing this out and have clarified this in the following sections.

**Changes:**

We have updated the beginning of the “**Results**” section to explicitly state which fMRI sessions we
used (page 4, lines 102 to 105):

“... Identification was performed between rs-fMRI data from day 1 and day 2 of HCP-YA
data and for each day FC was averaged across phase encoding directions. For prediction the FC
obtained on both days were averaged, resulting in one FC matrix per subject per co-fluctuation bin.
...”

Similarly, we have adjusted the “**Materials and Methods**” sub-section “**Datasets –**
**Human Connectome Project**” (page 18, lines 366 to 377) to avoid any ambiguity about which
sessions we used:

“... HCP Young-Adult (HCP-YA) . We used data obtained from four resting-state fMRI (rs-
fMRI) sessions taken from the HCP-YA S1200 release³³. Subjects were selected if data was
available for all four resting state sessions and 25 predefined behavioural variables of interest. This
resulted in a dataset consisting of 771 subjects (384 female, 387 male). Participants’ age ranged
from 22 to 37 ($M=28.41$, $SD=3.74$). The four sessions of rs-fMRI were obtained on two separate
172 days (each lasted ca. 15 minutes; ~60 minutes across all four sessions). On each day, two
sessions were recorded for different phase encoding directions (left-right [LR] and right-left [RL])
providing four overall rs-fMRI datasets. Scans were acquired using a 3T Siemens connectome-
Skyra scanner with a gradient-echo EPI sequence (TE=33.1ms, TR=720ms, flip angle = 52°,
2.0mm isotropic voxels, 72 slices, multiband factor of 8). ...”

We have also specified this in the “**Subject Specificity: Differential Identifiability and**
**Identification Accuracy**” sub-section of the “**Materials and Methods**” section (page 23, lines
482 to 487):

“... Identification was performed between rs-fMRI data from day 1 and day 2 of HCP-YA
data collection and for each day FC was averaged across phase encoding directions. That is, for
identification accuracy, identification was performed with day 1 data as a source and day 2 data as
a target, and vice versa resulting in two identification accuracy scores. These two scores were
averaged to obtain one overall identification accuracy. Differential identifiability was also estimated
between rs-fMRI data from day 1 and day 2. ...”

In addition, we have also adjusted a sentence in the “**Prediction of Behavioural and**
**Demographic Measures**” sub-section of the “**Materials and Methods**” section (page 23, lines
498 to 500):

“... Prediction was performed with FC matrices obtained using the three distinct strategies
outlined above and averaged across all four HCP resting state sessions with unique edges serving
as features. ...”

**2. Minor Comment:**

*Captions of some figures, such as figures 1 and 5 need to be completed. The captions are not*
*sufficiently provided information about how the figures were obtained and what they illustrated.*

**Response:**

We thank the reviewer for their comment about the captions of figures 1 and 5, and agree that they
were somewhat sparse. We have revised the captions to provide more detailed information on how
the figures were obtained and what they illustrate. Additionally, we have updated figures 3, 4, 6, 7,
and 8 to ensure they provide a more comprehensive understanding of our results. We have also
changed all the bar plots to box plots following Communications Biology guidelines.

**Changes:**

We have adjusted the caption for Figs. 1, 3, 4, 5, 6, 7, and 8 as follows:

Figure 1. Differential identifiability and identification accuracy in the HCP-YA dataset for the sequential sampling strategy (a and b), the individual bins sampling strategy (c and d), and the combined bins strategy (e). In a) and b), Threshold refers to the percentage of highest (HACF) or lowest (LACF) co-fluctuation frames chosen to estimate FC. In (e), the lower triangle shows differential identifiability, whereas the upper triangle shows identification accuracy achieved by each pair of combined bins. In each identification experiment subjects were resampled with replacement 1000 times, and in each resampling run only the subset of unique subjects were chosen to perform the identification analysis. Asterisks (“*”) in a) and b) indicate a significant difference as determined by a Wilcoxon-signed rank test between HACF-derived FC and LACF-derived FC at the .05 alpha significance level after Bonferroni correction and the error bars indicate a 95% confidence interval. In c) and d) the box plot indicates the median (center line) and the interquartile range.

Figure 3. Prediction scores (Pearson's r between observed and predicted on the y-axis) for 9 phenotypic targets averaged across the ten folds in the grouped cross-validation scheme when using FC estimates derived from time points at different levels of co-fluctuation magnitude using the sequential sampling strategy. "Threshold" refers to the percentage of highest (HACF) or lowest (LACF) co-fluctuation frames chosen to estimate FC. Upper and lower boundary of the fill colours indicate the standard deviation across folds. A threshold of 100% corresponds to full FC. These 9 targets are displayed, because they yielded best prediction accuracy using full FC compared to other targets displayed in the supplementary information.

Figure 4. Age prediction (Pearson's r) and sex classification (accuracy) using the sequential sampling strategy (**a and b**), individual bins (**c and d**), and combined bins (**e**) sampling strategies. In **a**) and **b**), "Threshold" (x-axis) refers to the percentage of highest (HACF) or lowest (LACF) co-fluctuation frames chosen to estimate FC and the y-axis indicates the prediction score. Upper and lower boundary of the fill colours indicate the standard deviation across folds. In **c**) and **d**) the box plot indicates the median (center line) and the interquartile range. In **e**) comparison operators indicate whether scores obtained by a co-fluctuation bin are equivalent to scores obtained by full FC (" $=$ ") or whether they are less (" $<$ ") or greater (" $>$ ") than scores obtained by full FC according to a 5% Bayesian ROPE⁴³.

Figure 5. Differential identifiability and identification accuracy in the HCP-A dataset for the sequential sampling strategy (a and b), the individual bins sampling strategy (c and d), and the combined bins strategy (e). In a) and b), “Threshold” refers to the percentage of highest (HACF) or lowest (LACF) co-fluctuation frames chosen to estimate FC. In (e), the lower triangle shows differential identifiability, whereas the upper triangle shows identification accuracy achieved by each pair of combined bins. In each identification experiment subjects were resampled with replacement 1000 times, and in each resampling run only the subset of unique subjects were chosen to perform the identification analysis. Asterisks (“*”) in a) and b) indicate a significant difference as determined by a Wilcoxon-signed rank test between HACF-derived FC and LACF-derived FC at the .05 alpha significance level after Bonferroni correction. The error bars indicate a 95% confidence interval. In c) and d) the box plot indicates the median (center line) and the interquartile range.

Figure 6. Prediction accuracy (Pearson's r) for four cognitive targets in the HCP-A dataset using the individual and combined bins strategy (**a**; individual bins are on the diagonal) and the sequential sampling strategy (**b**). Upper and lower boundary of the fill colours indicate the standard deviation across repeats. Age prediction (Pearson's r) and sex classification (balanced accuracy) using the sequential (**c** and **d**), individual bins (**e** and **f**), and combined bins (**g**) sampling strategies using the HCP-A sample. In **b**, **c** and **d**, "Threshold" (x-axis) refers to the percentage of highest (HACF) or lowest (LACF) co-fluctuation frames chosen to estimate FC and the y-axis displays the prediction scores. In **e** and **f** the box plot indicates the median (center line) and the interquartile range. In **g** comparison operators indicate whether scores obtained by a co-fluctuation bin are equivalent to scores obtained by full FC ("=") or whether they are less ("<") or greater (">") than scores obtained by full FC according to a 5% Bayesian ROPE.

Figure 7. Correlations between SC and FC averaged across subjects in the HCP-YA sample for **a)** individual (on-diagonal) and combined bins (off-diagonal) and the **b)** sequential paradigm. In **b)** upper and lower bounds of fill colours indicate minimum and maximum correlations across subjects. “Threshold” (x-axis) refers to the percentage of highest (HACF) or lowest (LACF) co-fluctuation frames chosen to estimate FC and the y-axis shows the Pearson correlation between FC edges and SC edges.

Figure 8. Prediction accuracy for three cognitive targets in the HCP-YA sample (Pearson's r between observed and predicted targets) when removing SC from FC estimates for each subject in the combined and individual (a) and the sequential (b) sampling strategies. Age prediction (Pearson's r) and sex classification (accuracy) using the sequential (c and d), individual bins (e and f), and combined bins (g) sampling strategies after removal of SC from FC. In b), c) and d), "Threshold" refers to the percentage of highest (HACF) or lowest (LACF) co-fluctuation frames chosen to estimate FC. In e) and f) the box plot indicates the median (center line) and the interquartile range. In g) comparison operators indicate whether scores obtained by a co-fluctuation bin are equivalent to scores obtained by full FC (" $=$ ") or whether they are less (" $<$ ") or greater (" $>$ ") than scores obtained by full FC according to a 5% Bayesian ROPE.

**3. Minor Comment:**

*Before resubmitting, I would suggest the authors carefully go through the article and break the long*
*sentences into short and clear sentences.*

**Response:**

We have carefully reviewed the article and made efforts to reduce the complexity and
length of sentences where appropriate. We recognize the importance of clear and concise writing,
and have made a concerted effort to ensure that our writing is easy to follow and understand. We
appreciate your suggestion and are grateful for the opportunity to improve the readability of our
manuscript.

Referee #2

*Sasse and colleagues study temporal fluctuations in FC within the duration of a functional*
*magnetic resonance imaging (fMRI) scanning session. Employing three time frame sampling*
*strategies, we investigated predictiveness of targets across the domains of cognition, behaviour,*
*personality, and demographics from FC including all frames. Given the widespread use of*
*functional connectivity measures as features for predictive models in neuroimaging, this paper can*
*make a valuable contribution to the research literature. This reviewer recommends integration of*
*the following action points:*

**Major Comments:**

**1. Major Comment:**

*Introduction: first paragraph appears quite general and could get faster at the research question at*
*hand.*

**Response:**

Thank you for your feedback on our introduction. We appreciate your comments and agree that the
focus should be on the specific research question as early as possible. However, we decided to
provide a general introduction to give appropriate context to the research topic in particular for a
wider readership with basic or no knowledge on this topic. We have now added a sentence to the
paragraph to provide more information on the research question. Specifically, we have highlighted
that recent research has shifted its focus towards investigating how different time points in the rs-
fMRI time series contribute to some of the properties of FC.

**Changes:**

We have adjusted the first paragraph as follows (page 2, lines 28 to 39):

*“... In an effort to understand how brain organisation facilitates flexible yet specialised*
*cognitive function, much neuroscientific research has focused on the functional connectivity (FC)*
*between brain areas by investigating their pairwise correlations of functional magnetic resonance*
*imaging (fMRI) blood oxygen level-dependent (BOLD) signals¹⁻³. These pairwise correlations are*
*assumed to represent the strength of connectivity (also called edges) between brain areas (also*
*called nodes). Notably, considerable promise for the eventual applicability of FC as a biomarker*
*has been shown with numerous studies demonstrating that FC differs between individuals, is*
*stable within an individual⁴⁻⁸, and relates to individual-level cognition^{4,9-13} as well as clinically*
*relevant symptoms of mental disorders¹⁴⁻¹⁷. In this context, recent research has focused on*
*investigating how different time points in the rs-fMRI time series contribute to some of these*

properties of FC¹⁸⁻²⁰. However, it is not yet well understood how different time points contribute
specifically to the predictive utility of FC. ...”

**2. Major Comment:**

*Introduction / "These time points of high-amplitude co-fluctuations, called “events”, contribute*
*disproportionately to FC and are thought to reflect fluctuations in cognitive state”: not all high-*
*magnitude changes in resting state signals are thought to be cognitively or behaviourally relevant;*
*please smoothen statement by integrating:*

1. Uddin, L. Q. (2020). *Bring the noise: reconceptualizing spontaneous neural activity. Trends*
*in Cognitive Sciences, 24(9), 734-746.*

2. Li, J., Bolt, T., Bzdok, D., Nomi, J. S., Yeo, B. T., Spreng, R. N., & Uddin, L. Q. (2019).
*Topography and behavioral relevance of the global signal in the human brain. Scientific*
*reports, 9(1), 1-10.*

3. Bzdok, D., Varoquaux, G., Grisel, O., Eickenberg, M., Poupon, C., & Thirion, B. (2016).
*Formal models of the network co-occurrence underlying mental operations. PLoS*
*computational biology, 12(6), e1004994.*

**Response:**

Thank you for your comment and for suggesting these articles. We appreciate the crucial context
and insight they provide for our research topic. We agree that there is some debate about the
cognitive or behavioural relevance of high-magnitude changes in resting state signals. As
highlighted in recent reconceptualizations of spontaneous neural activity, such as those proposed
by Uddin (2020) and Li, Bolt, et al. (2019), changes in resting state signals, for example the global
signal, may not always be thought of as related to cognitive processing. Although there is evidence
pointing towards their significance in terms of brain organisation (Bolt et al., 2022; Bzdok et al.,
2016) more work is needed to clearly understand these signals’ aetiology and how they underpin
cognition and behaviour if at all (Gonzalez-Castillo, 2022).

**Changes:**

In order to smoothen the statement about cognitive and behavioural significance of resting state
fMRI signal changes over time, we have integrated and cited the suggested articles as follows
(page 2, lines 49 to 54):

“ ... Despite the high magnitude of changes in resting state fMRI signals, the extent to
which they contribute to cognition and behaviour is still a matter of debate in the literature²³⁻²⁵.
Although approaches to decompose task activity into underlying recruitment of resting state
networks have been proposed, further research is needed to investigate how resting state signals

and their changes over time underlie cognitive and behavioural performance²⁶. ...”

**3. Major Comment:**

*Methods: why and how were the 2 resting-state (out of 4) from the HCP data selected? Perhaps it*
*is wording, different phase encoding acquisitions could be different sessions, too.*

**Response:**

We thank the reviewer for pointing this out and have clarified this in the following sections. Please
note that minor comment 1 from Referee 1 points to the same issue. To avoid the reviewers having
to go back and forth, the response is copied below.

**Changes:**

We have updated the beginning of the “**Results**” section to explicitly state which fMRI sessions we
used (page 4, lines 102 to 105):

“... Identification was performed between rs-fMRI data from day 1 and day 2 of HCP-YA data
collection and for each day FC was averaged across phase encoding directions. For prediction the
FC obtained on both days were averaged, resulting in one FC matrix per subject per co-fluctuation
bin. ...”

Similarly, we have adjusted the “**Materials and Methods**” sub-section “**Datasets –**
**Human Connectome Project**” (page 18, lines 366 to 377) to avoid any ambiguity about which
sessions we used:

“... *HCP Young-Adult (HCP-YA)*. We used data obtained from four resting-state fMRI (rs-
fMRI) sessions taken from the HCP-YA S1200 release³³. Subjects were selected if data was
available for all four resting state sessions and 25 predefined behavioural variables of interest. This
resulted in a dataset consisting of 771 subjects (384 female, 387 male). Participants’ age ranged
from 22 to 37 ($M=28.41$, $SD=3.74$). The four sessions of rs-fMRI were obtained on two separate
308 days (each lasted ca. 15 minutes; ~60 minutes across all four sessions). On each day, two
sessions were recorded for different phase encoding directions (left-right [LR] and right-left [RL])
providing four overall rs-fMRI datasets. Scans were acquired using a 3T Siemens connectome-
Skyra scanner with a gradient-echo EPI sequence (TE=33.1ms, TR=720ms, flip angle = 52°,
2.0mm isotropic voxels, 72 slices, multiband factor of 8). ...”

We have also specified this in the “**Subject Specificity: Differential Identifiability and**
**Identification Accuracy**” sub-section of the “**Materials and Methods**” section (page 23, lines
482 to 487):

“... Identification was performed between rs-fMRI data from day 1 and day 2 of HCP-YA

data collection and for each day FC was averaged across phase encoding directions. That is, for
identification accuracy, identification was performed with day 1 data as a source and day 2 data as
a target, and vice versa resulting in two identification accuracy scores. These two scores were
averaged to obtain one overall identification accuracy. Differential identifiability was also estimated
between rs-fMRI data from day 1 and day 2. ...”

In addition, we have also adjusted a sentence in the “**Prediction of Behavioural and**
**Demographic Measures**” sub-section of the “**Materials and Methods**” section (page 23, lines
498 to 500):

“... Prediction was performed with FC matrices obtained using the three distinct strategies
outlined above and averaged across all four HCP resting state sessions with unique edges serving
as features. ...”

**4. Major Comment:**

*Methods: more information should be provided on whether and how the global signal regression*
*was present and computed in the main analysis?*

**Response:**

Thank you for your comment and question. We confirm that global signal regression was
performed in the main analysis to reproduce Zamani Esfahlani et al. (2020)’s preprocessing as
closely as possible. We updated the subsection titled “**Image pre-processing**” in the “**Materials**
**and Methods**” section to provide more specific context and information, and include the exact
number of parameters for each noise component. In addition, we also provide results without
global signal regression in the supplementary. We have adjusted the manuscript accordingly as
indicated below.

**Changes:**

The changes that we applied on page 19, lines 398 to 402, are also displayed below for
convenience:

“For both datasets, we regressed out confounds, linearly detrended and bandpass filtered
the signal at 0.008 - 0.08 Hz using “nilearn.image.clean_img”. For the main analysis, this included
mean time courses of the white matter (WM), cerebro-spinal fluid (CSF), and global signal (GS), as
well as their squared terms, and the temporal derivatives of the mean signals as well as their
squared terms as confounds, resulting in 12 parameters (4 for each noise component). In the
supplementary we also provide results without global signal regression.”

**5. Major Comment:**

*Methods: more information could/should be provided on how the spike regression for signal*
*cleaning was carried out quantitatively.*

**Response:**

We have updated the “**Image pre-processing**” subsection of the “**Materials and Methods**”
section to better reflect this. We followed this approach for spike regression described in the
original publication on edge time series by Zamani Esfahlani et al., (2020) to ensure comparability
between our results and theirs. This approach allowed us to identify and remove the effect of
spikes in motion in a simple and straightforward manner, while maintaining the integrity of the
underlying signal.

**Changes:**

To clarify how spike regression was performed we have revised lines 402 to 404 on page 19 in the
“**Materials and Methods**” sub-section “**Image pre-processing**”:

“... A binary spike regressor was further added for each fMRI frame exceeding a motion
threshold (“1” where root mean squared framewise displacement [FD] > 0.25 mm, and “0” where
FD < 0.25 mm). ...”

**6. Major Comment**

*Methods / connectome prediction: where the features cleaned or encoded different? For what*
*reasons where the kernels picked in ridge regression modeling? Was any hyper parameter*
*selection done, and, if yes, how? How was the confounding removal integrated into the cross*
*validation workflow?*

**Response:**

We thank the reviewer for these detailed questions and comments, as we recognise the
importance of describing machine learning pipelines in as much detail as possible. The choice of
using ridge regression with a Pearson kernel in particular was based on prior literature suggesting
its efficacy for modelling data of similar nature (He et al., 2020; Li, Kong, et al., 2019; Ooi et al.,
2022). Our focus was to adhere to a standard, well-established method and direct our attention on
the contribution of individual co-fluctuation bins towards the information contained in FC rather than
extensive model tuning. In addition, the pipeline was designed to avoid over-optimistic out-of-
sample performance estimates and to avoid test-to-train leakage. Specifically, we took care to
perform a nested cross-validation for hyperparameter tuning (i.e. performing an inner cross-
validation to select the best hyperparameter which is then used to train the model and evaluate on

completely unseen test data). Similarly, confound removal was performed by training a confound
regression model on the training data only. That is, in each cross-validation split a confound
regression model was trained on only the training data in that particular split, which could then be
applied to the test data. We have adjusted the manuscript as shown below to discuss these steps
in more detail.

**Changes:**

To better address some of these questions in the manuscript, we have made important changes in
the “**Results**” section (pages 5 to 6, lines 127 to 140):

“... In order to test whether the differing behaviour of I_{Diff} and I_{Acc} can inform about the
predictive utility, we then applied the sampling strategies to predict phenotypes using kernel ridge
regression. We selected phenotypes used in ref.⁴² from the categories “Cognition”, “In-scanner task
performance”, and “Personality” (see Table S1). Results displayed here consist of the 9 phenotypic
targets with highest prediction accuracy based on the full FC. To assess the out-of-sample
prediction accuracy of our models in the HCP-YA dataset, we performed a 10-fold nested cross-
validation (CV) procedure. We ensured that family members were always kept within the same fold
in the 10-fold CV to maintain independence between folds. We used a 5-fold inner CV on the
training folds to select the hyperparameters for ridge regression (l2-regularisation strength) and for
support vector machines (C parameter) in a consistent manner. The best parameters were then
fitted on the training folds of the outer CV and tested on the outer CV test fold. Similarly, to avoid
test-to-train leakage during confound removal, we trained a confound regression model on the
training data only, to remove the effects of age, sex, and framewise displacement (for more
information see “Materials and Methods”, subsection “Prediction of Behavioural and Demographic
Measures”). ...”

We have also added some clarifications to the “**Materials and Methods**” sub-section
“**Prediction of Behavioural and Demographic Measures**” on page 23 (lines 507 to 511):

“... To control for confounding influences, age at scan, sex at birth and framewise
displacement (FD) of resting state fMRI recordings were regressed out from the targets in a CV-
consistent fashion as these have been found to correlate with behavioural variables^{12,55}. In this
context, CV-consistency means training the confound regression model on the training partition of
a split only, and applying this model to the test data subsequently, to avoid test-to-train leakage .
...”

and on page 24 (lines 524 to 534):

“... To assess out-of-sample prediction accuracy in the HCP-YA dataset, a 10-fold nested
cross-validation (CV) was performed for each FC representation. The folds were split such that
family members were always within the same fold, so that independence between folds was
maintained. To select the l2-regularisation strength for ridge regression and classification as well

as the C parameter for the support vector classifiers in CV-consistent fashion, we used a 5-fold
inner CV on the training folds. Candidate values for the l2-regularisation strength used in
hyperparameter tuning were: {0, 0.00001, 0.0001, 0.001, 0.004, 0.007, 0.01, 0.04, 0.07, 0.1, 0.4,
419 0.7, 1, 1.5, 2, 2.5, 3, 3.5, 4, 5, 10, 15, 20, 30, 40, 50, 60, 70, 80, 100, 150, 200, 300, 500, 700,
1000, 10000, 100000, 1000000}. Candidate values for the C parameter of the support vector
classifiers were 50 numbers between 0.01 and 100 evenly spaced on the log scale generated by
the “numpy.geomspace” function. ...”

**7. Major Comment:**

*General: the manuscript appears to be meta-stable between 'identification' and 'identifiability',*
*which are similar, but different notions - please keep and argue for one of them throughout the*
*whole work.*

**Response:**

Thank you for your feedback regarding our use of the term "identifiability" in the manuscript. We
appreciate you pointing out the ambiguity and slight inconsistency in our use of this term. We
acknowledge that we often used "subject identifiability" when we meant "subject specificity", which
underlies both of the metrics "differential identifiability" and "identification accuracy". We do not
want to confuse "subject specificity" and "differential identifiability". We recognize the importance of
distinguishing between identification accuracy and differential identifiability as well as the concept
of subject specificity that underlies both of these metrics. This is particularly important given that
based on our findings, these metrics can behave quite differently on the same data. Thus, we have
adjusted the manuscript to use “subject specificity” when we refer to the underlying more general
concept that underlies both metrics, so that we only use the terms “identification accuracy” and
“differential identifiability” when we refer to the actual metrics.

**Changes:**

We have carefully reviewed our manuscript and replaced “subject identifiability” with “subject
specificity” where necessary to improve the precision of our terminology in the manuscript. For
example, we have adjusted the title of the manuscript to: “Intermediately Synchronised Brain
States optimise Trade-off between Subject Specificity and Predictive Capacity”. In addition, there
are a number of places in the main text where we have replaced “subject identifiability” with
“subject specificity” when we are writing about the general concept of subject specificity rather than
any of the individual metrics (“identification accuracy” or “differential identifiability”). For example in
the introduction, page 3, lines 62-64: “HACF-derived FC has been shown to yield enhanced
subject specificity, bringing up the question whether using these HACF time frames might also
amplify brain-behaviour correlations^{18,19,29}.” Another example can be found at the beginning of the

discussion (page 14, lines 235-236): "It has been suggested that the use of HACF frames with
enhanced subject specificity may amplify brain-behaviour associations^{20,29}."

**Minor Comments:**

**1. Minor Comment:**

*Better to avoid abbreviations in subtitles.*

**Response:**

Thank you for your comment. We have removed the abbreviations from the subtitles in the revised
version of the manuscript to improve clarity and comprehension.

**2. Minor Comment**

*Figures: the 'threshold' axis may be hard to follow for several readers and could be explained*
*more. Also the nature of the metric on the y axis is not always directly apparent.*

**Response:**

Thank you for your constructive feedback regarding the figures in our manuscript. We agree that
the "threshold" axis might be challenging to follow for some readers, and the metric on the y-axis
might not always be apparent. We have updated the relevant figure captions to explain more
clearly what we mean by "Threshold" in the context of the sequential sampling strategy. We have
also updated them to clearly specify the metrics on the y-axis. Please see the response to the
second minor comment by the first reviewer (**Referee #2, 2. Minor Comment**), where all relevant
figures and their captions have been updated and displayed.

**References**

- Benavoli, A., Corani, G., Demšar, J., & Zaffalon, M. (2017). Time for a Change: A Tutorial for
Comparing Multiple Classifiers Through Bayesian Analysis. *Journal of Machine Learning*
*Research*, 18(77), 1–36.
- Bolt, T., Nomi, J. S., Bzdok, D., Salas, J. A., Chang, C., Thomas Yeo, B. T., Uddin, L. Q., &
Keilholz, S. D. (2022). A parsimonious description of global functional brain organization in
three spatiotemporal patterns. *Nature Neuroscience*, 25(8), Article 8.
<https://doi.org/10.1038/s41593-022-01118-1>
- Bzdok, D., Varoquaux, G., Grisel, O., Eickenberg, M., Poupon, C., & Thirion, B. (2016). Formal
Models of the Network Co-occurrence Underlying Mental Operations. *PLoS Computational*
*Biology*, 12(6), e1004994. <https://doi.org/10.1371/journal.pcbi.1004994>
- Gonzalez-Castillo, J. (2022). Traveling and standing waves in the brain. *Nature Neuroscience*,
25(8), 980–981. <https://doi.org/10.1038/s41593-022-01119-0>
- He, T., Kong, R., Holmes, A. J., Nguyen, M., Sabuncu, M. R., Eickhoff, S. B., Bzdok, D., Feng, J.,
& Yeo, B. T. T. (2020). Deep neural networks and kernel regression achieve comparable
accuracies for functional connectivity prediction of behavior and demographics.
*NeuroImage*, 206, 116276. <https://doi.org/10.1016/j.neuroimage.2019.116276>
- Li, J., Bolt, T., Bzdok, D., Nomi, J. S., Yeo, B. T. T., Spreng, R. N., & Uddin, L. Q. (2019).
Topography and behavioral relevance of the global signal in the human brain. *Scientific*
*Reports*, 9(1), Article 1. <https://doi.org/10.1038/s41598-019-50750-8>
- Li, J., Kong, R., Liégeois, R., Orban, C., Tan, Y., Sun, N., Holmes, A. J., Sabuncu, M. R., Ge, T., &
Yeo, B. T. T. (2019). Global signal regression strengthens association between resting-
state functional connectivity and behavior. *NeuroImage*, 196, 126–141.
<https://doi.org/10.1016/j.neuroimage.2019.04.016>
- More, S., Eickhoff, S. B., Caspers, J., & Patil, K. R. (2021). Confound Removal and Normalization
in Practice: A Neuroimaging Based Sex Prediction Case Study. *Machine Learning and*
*Knowledge Discovery in Databases. Applied Data Science and Demo Track*, 12461, 3–18.
https://doi.org/10.1007/978-3-030-67670-4_1
- Ooi, L. Q. R., Chen, J., Zhang, S., Kong, R., Tam, A., Li, J., Dhamala, E., Zhou, J. H., Holmes, A.
498 J., & Yeo, B. T. T. (2022). Comparison of individualized behavioral predictions across
anatomical, diffusion and functional connectivity MRI. *NeuroImage*, 263, 119636.
<https://doi.org/10.1016/j.neuroimage.2022.119636>
- Snoek, L., Miletić, S., & Scholte, H. S. (2019). How to control for confounds in decoding analyses
of neuroimaging data. *NeuroImage*, 184, 741–760.
<https://doi.org/10.1016/j.neuroimage.2018.09.074>

- Uddin, L. Q. (2020). Bring the Noise: Reconceptualizing Spontaneous Neural Activity. *Trends in*
*Cognitive Sciences*, 24(9), 734–746. <https://doi.org/10.1016/j.tics.2020.06.003>
- Zamani Esfahlani, F., Jo, Y., Faskowitz, J., Byrge, L., Kennedy, D. P., Sporns, O., & Betzel, R. F.
(2020). High-amplitude cofluctuations in cortical activity drive functional connectivity.
*Proceedings of the National Academy of Sciences of the United States of America*,
117(45), 28393–28401. <https://doi.org/10.1073/pnas.2005531117>

REVIEWERS' COMMENTS:

Reviewer #1 (Remarks to the Author):

In my previous review, I provided several comments to enhance the quality and clarity of the manuscript. The authors have made substantial revisions and addressed these concerns adequately. I would like to commend the authors for their thorough revisions and for addressing the concerns raised in my previous review. The changes made have significantly strengthened the manuscript.

Reviewer #2 (Remarks to the Author):

The authors have done a great job addressing my previous reviewer comments.

Danilo Bzdok, MD, PhD